# An observational estimate of Arctic UV-absorbing aerosol direct radiative forcing on instantaneous and climatic scales

Blake T. Sorenson[1*], Jianglong Zhang[1], Jeffrey S. Reid[2], and Peng Xian[2]

[1]Department of Atmospheric Sciences, University of North Dakota, Grand Forks, North Dakota, 58202, United States of America
[2]Marine Meteorology Division, Naval Research Laboratory, Monterey, California, 93943, United States of America
[*]Now at the National Research Council, Monterey, California, 93943, United States of America

*Correspondence to:* Blake T. Sorenson (blake.sorenson@und.edu)

**Abstract.** Using co-located satellite observations from the Aqua Moderate resolution Imaging Spectroradiometer, the Aqua Cloud and the Earth Radiant Energy System, the Special Sensor Microwave Imager / Sounder, and the Ozone Monitoring Instrument, we investigated changes in absorbing aerosol direct radiative forcing (ADRF) in the spring through fall Arctic from 2005 – 2020 through an observation based method, assisted by a neural network for estimating cloud and aerosol free sky Top-of-Atmosphere (TOA) radiative fluxes, and an innovative, Monte-Carlo-based method for estimating uncertainties in derived ADRF values. This study suggests that Arctic ADRF is a strong function of observing conditions, and changes in Arctic sea ice concentration (SIC) and cloud properties introduce a complex scenario for estimating ADRF. For example, the TOA ADRF reverses sign from negative (cooling) to positive (warming) for SIC above 60% for a region with a relatively cloud free scene. ADRF trends over Arctic land surfaces are primarily negative. Strong negative ADRF trends of up to -4 Wm$^{-2}$ were found over northern Russia and northern Canada in the summer months. Both positive and negative ADRF trends were found over the Arctic Ocean in the boreal summer, though these trends are much weaker than the over-land trends. Positive ADRF trends in the Arctic Ocean north of northeastern Russia and northern Canada are as high as +1.0 Wm$^{-2}$ per study period. The trend results suggest that increasing amounts of absorbing aerosols in the Arctic have a cooling effect from TOA that could act to counter Arctic warming.

## 1. Introduction

The Arctic is a complex and changing region, especially due to recent drastic decreases in summertime Arctic sea ice coverage (Comiso, 2012; Kwok and Rothrock, 2009). Warming in the Arctic over the past few decades has been much stronger than the global average, with this phenomenon being referred to as "Arctic Amplification" (Dai et al., 2019; Serreze and Barry, 2011; Serreze and Francis, 2006). As the Arctic warms, bright ice- and snow-covered surfaces are converting to darker ocean and land surfaces, increasing the amount of absorbed solar energy and fueling further warming and ice melt (Dai et al., 2019; Kashiwase et al., 2017; Perovich et al., 2007). One factor complicating the changing Arctic is the intrusion of significant aerosol plumes, primarily biomass burning (BB) smoke from lower latitudes, into the Arctic region. Such intrusions of BB smoke into the Arctic region have become more frequent over

the past two decades (Sorenson et al., 2023; Xian et al., 2022a, b). Aerosol particles are well known to impact the climate directly through absorption and scattering of shortwave solar radiation and absorption of earth emitted longwave radiation. Indirectly, aerosol particles affect the climate through their interactions with clouds. Aerosol

particles can act as cloud condensation nuclei, leading to the formation of smaller cloud droplets that increase cloud albedo (Twomey, 1977) and affect cloud lifetime (Albrecht, 1989). Over the Arctic region, light-absorbing aerosol particles can also be deposited on snow- and ice-covered surfaces (e.g. Khan et al., 2023), reducing surface reflectivity and accelerating snow/ice melt, causing a positive (i.e., warming) radiative forcing (Flanner et al., 2007; Hansen and Nazarenko, 2004).

While it is well documented that the Arctic climate is sensitive to aerosol particles (Feng et al., 2013; Flanner, 2013; Samset et al., 2013; Shindell and Faluvegi, 2009), with the detectable increases in aerosol events over the Arctic regions for the past two decades (Sorenson et al., 2023; Xian et al., 2022a, b), it is necessary to carefully quantify the impact of aerosol particles on Arctic climate. Many studies have investigated aerosol-climate impacts in the Arctic region, primarily through the use of numerical climate models and/or aerosol analyses (Breider et al., 2017;

DeRepentigny et al., 2022; Feng et al., 2013; Markowicz et al., 2017, 2021; Oshima et al., 2020; Schacht et al., 2019). Similarly, previous studies have investigated the interactions between aerosol particles and snow- and ice-covered surfaces, with many using global climate models to determine how the deposition of absorbing particles onto sea ice and snow impacts the aerosol-radiation interactions (Bond et al., 2013; Flanner et al., 2007; Gagné et al., 2015; Schacht et al., 2019; Shindell and Faluvegi, 2009). Some studies have even investigated how changes in sea ice coverage affect

aerosol radiative forcing in the Arctic. Using a global climate model, Struthers et al. (2011) found that reductions in Arctic sea ice extent led to increased emissions of sea spray/salt aerosol particles, with the associated increase in total aerosol optical depth leading to stronger aerosol radiative cooling effects and a negative feedback on the Arctic climate.

While numerical models have been used extensively to study the impact of aerosol particles on Arctic climate, the

observation-based study of Arctic aerosol-climate impact, which can prove valuable for evaluation of model-based studies, remains a very challenging research topic. Observing Arctic aerosol particles from traditional, passive-based sensors such as the MODerate resolution Imaging Spectroradiometer (MODIS) or the Visible Infrared Radiometer Suite (VIIRS) is difficult due to the bright ice and snow surfaces that frequently cover the Arctic (Martin, 2008). Further, active-based sensors such as the Cloud-Aerosol Lidar with Orthogonal Polarization (CALIOP) have much

smaller fields of view than passive imagers, have orbits that only extend to 82° N and miss a large part of the Arctic, and are also at times sensitive to reduced signal to nose over bright surfaces (Toth et al., 2018).

Attempts have been made in recent years to detect aerosol features over the bright surfaces in the Arctic from passive-based satellite sensors, with methods developed using combined Aqua and Terra MODIS data (Mei et al., 2013a) and others with observations from the Advanced Along-Track Scanning Radiometer (AATSR) (Mei et al., 2013b, 2020;

Swain et al., 2024). These methods have limitations, though, with the AATSR-based method focusing on coarse-mode aerosol particles. Additionally, Arctic aerosol retrievals from those methods have data records that are too short for a long-term trend analysis (e.g. Mei et al., 2013a), and/or are without a companion sensor providing broadband

observations enabling observation-based ADRF study (e.g. Swain et al., 2024). As an alternative, the Ozone Monitoring Instrument (OMI) ultraviolet aerosol index (UVAI), through detection of UV-absorbing aerosols by comparing observed radiance and computed radiance assuming a Rayleigh atmosphere at the 354 nm channel, is able to detect aerosols over bright surfaces such as desert and cloud and over bright snow- and ice-covered surfaces (Alfaro-Contreras et al., 2014, 2016; Hsu et al., 1999; Torres et al., 2012). Furthermore, with the combined use of OMI UVAI and broadband observations from the Cloud and the Earth Radiant Energy System (CERES), Feng and Christopher (2015) studied the direct radiative effect of BB aerosols over marine stratocumulus clouds, further showing the utility of OMI UVAI measurements to study aerosol radiative forcing over bright surfaces. Recent work has even demonstrated how the OMI UVAI parameter may be used to study instantaneous and climatological Arctic aerosol patterns over both dark and bright surfaces (Sorenson et al., 2023; Zhang et al., 2021). Thus, with the combined use of observations from OMI and Aqua CERES, which are both included in the A-train constellation and have near coinciding observations within 30 minutes, it is feasible to quantify absorbing aerosol direct radiative forcing (ADRF) from an observation-based analysis.

Quantifying ADRF from observations, while feasible, is nevertheless daunting. Frequent and significant changes in surface properties due to the retreat and expansion of sea ice make the Arctic a uniquely difficult region to study the ADRF from observations. In addition to decreasing sea ice, observation-based studies also found increases in Arctic summertime cloud cover over the last few decades on the order of 10% per decade (Abe et al., 2016; Philipp et al., 2020; Schweiger, 2004; Schweiger et al., 2008), adding another layer of complexity to observational-based aerosol forcing analyses. The impact of sea ice change and the behavior of Arctic clouds on the radiative effect of an aerosol plume in the Arctic can be seen in Fig. 1. Aqua MODIS true color imagery (Fig. 1a) and OMI UVAI (Fig. 1Figure 1b) reveal a plume of BB smoke from central Russia that extends north from the mainland, over the exposed Arctic Ocean water and eventually over the sea ice. The Aqua CERES TOA shortwave flux (SWF) measurements (Fig. 1c and with OMI UVAI overlaid in Fig. 1d) within the plume region over the ocean water exhibit higher upwelling SWF than in the surrounding regions over the water. In this case, the aerosols have a brightening effect, causing more upwelling TOA radiation than in clear-sky regions. In the second case, however, Aqua MODIS true color imagery (Fig. 1Figure 1e) and OMI UVAI (Fig. 1f) show a dense smoke plume over northeastern Russia and extending north over both Arctic sea ice and cloud. The visible imagery and CERES SWF measurements (Fig. 1g and with OMI UVAI overlaid in Fig. 1h) show that the same smoke plume has both a darkening and a brightening effect: brightening over the landmass of northeastern Russia and darkening over the sea ice. These two cases illustrate the complex factors that affect Arctic aerosol radiative effects.

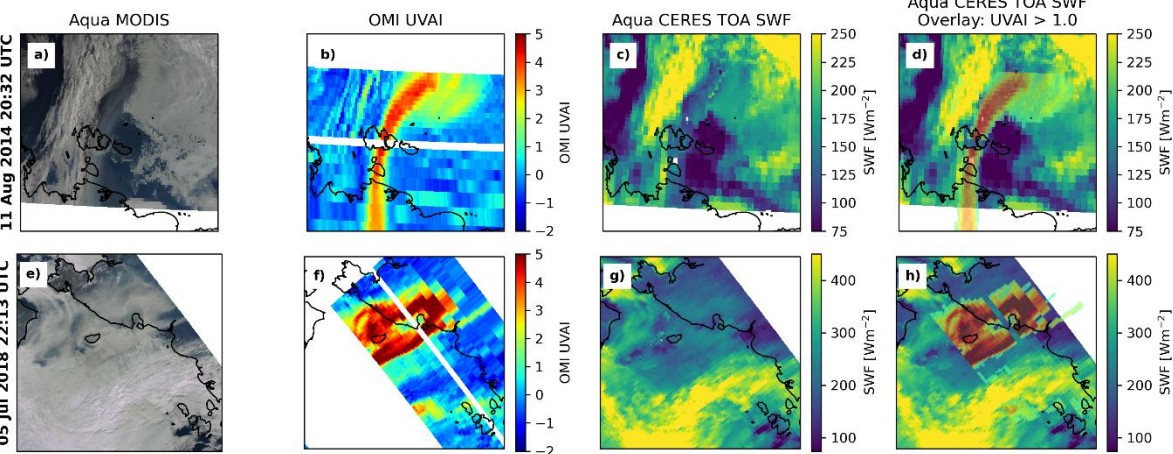

**Figure 1.** Comparison of the radiative effect of Arctic biomass burning smoke plumes of a smoke plume over ocean water from the 22:11 UTC 11 August 2014 OMI swath (top row), and of a smoke plume over ice and snow-free land from the 22:13 UTC 5 July 2018 OMI swath (bottom row). First column: Aqua MODIS true-color image. Second column: OMI UV aerosol index. Third column: Aqua CERES top of atmosphere (TOA) shortwave flux (SWF). Fourth column: CERES TOA SWF with OMI UVAI overlaid (fourth column).

With OMI UVAI being capable of detecting UV-absorbing aerosols over snow, ice, and clouds, the data provide a unique pathway for studying the complicated ADRF in the Arctic. In this study, we seek to use OMI UVAI and colocated CERES observations to derive a first-of-its-kind, observational estimate of absorbing aerosol direct radiative forcing trends in the Arctic. This analysis focuses on the radiative forcing of absorbing aerosols, so the radiative cooling impacts of sulfates or other scattering aerosol particles are not included in this study. Additionally, as this study focuses on only the direct radiative impacts of absorbing aerosols, we do not include the impacts of aerosol-cloud interactions or the radiative impacts of aerosol-cryosphere interactions (such as the deposition of absorbing aerosol particles onto bright snow- and ice-covered surfaces). In Section 2, we describe the data sources and variables analyzed in this study. In Section 3, we develop methods for estimating aerosol-free TOA upwelling SWF in smoky regions using a neural network-based approach. In Section 4, we estimate observation-based, long-term trends in ADRF using a look-up table (LUT) of aerosol forcing properties and applied Monte Carlo simulations for estimating uncertainties in the trend analyses.

## 2. Data

To perform an observational study of ADRF in the Arctic, observations of aerosol loading, TOA upwelling shortwave flux, surface type (including sea ice concentration (SIC)), and cloud condition are needed. Satellite-based sensors from the A-train constellation provide observations of the needed atmospheric variables (aerosol loading proxied by UVAI from OMI, shortwave flux from Aqua CERES, and cloud conditions from Aqua MODIS) within 15 minutes of each other (Schoeberl, 2002). While not part of the A-train constellation, SSMIS daily SIC retrievals can provide surface

type information for the analysis. The long data record of the A-train satellite sensors and the SSMIS instruments allow for a long-term analysis of Arctic ADRF.

## 2.1 OMI UV Aerosol Index Data

The Ozone Monitoring Instrument (OMI), onboard the Aura satellite, measures reflected solar energy from the ultraviolet (UV) to the visible spectrum (270 nm – 500 nm) (Levelt et al., 2006). Aura orbits the Earth in a sun-synchronous orbit at 705 km of altitude, an orbital inclination of 98.2º, and an equatorial crossing time of ~1:45 PM local time. In this study, UV-absorbing Arctic aerosol particles are detected using OMI ultraviolet aerosol index (UVAI) data, which relates the observed UV radiance at 354 nm to a calculated UV radiance assuming a purely
Rayleigh atmosphere using equation 1:

$$UVAI = -100 \log \left[\frac{I_\lambda^{obs}}{I_\lambda^{cal}}\right] \tag{1}$$

where $I_\lambda^{obs}$ is the observed 354 nm radiance and $I_\lambda^{cal}$ is the calculated 354 nm radiance assuming a Rayleigh atmosphere. Level 2 OMI OMAERUV V003 data from April to September of each year from 2005 through 2020 were obtained from the Goddard Earth Science (GES) Data and Information Services Center (DISC) archive (Torres, 2006).

Arctic OMI UVAI data needed to be extensively quality controlled and corrected to enable a study as is presented here. We followed the methods described by Sorenson et al. (2023), where the raw UVAI data are converted to Arctic UVAI perturbations relative to a climatological UVAI that is binned by viewing geometry and surface condition, thus removing substantial viewing geometry and surface condition-related uncertainties in the Arctic UVAI data. The OMI sensor has also suffered from the row anomaly problem, a dynamic and changing problem in which certain sensor
rows become contaminated and unusable, since 2007 (Torres et al., 2018). The number of contaminated rows varied from 2007 to the present, with about 50% of the OMI rows currently being contaminated, so we applied the row anomaly quality control flag in the OMI dataset to exclude all flagged, row anomaly-affected rows from our analysis. Sorenson et al. (2023) also identified additional OMI sensor rows in the data record that are affected by the row anomaly problem in the Arctic but are not flagged accordingly in the L2 OMAERUV data files; in this analysis, those
additional unflagged, contaminated rows were excluded from this analysis.

Other satellite sensors provide measurements of UVAI, including the Tropospheric Monitoring Instrument (TROPOMI), onboard Sentinel-5p (Veefkind et al., 2012). TROPOMI has significantly higher resolution than OMI (3.5 km x 7 km nadir pixel size for TROPMI and 13 km x 24 km nadir pixel size for OMI) and does not suffer from row anomaly issue, but we do not include TROPOMI data in this study for several reasons. First, the data record for
TROPOMI does not extend as far back as the OMI data record, with Sentinel 5-p being launched in 2017 and Aura being launched in 2004. Second, spatially and temporally colocated TROPOMI and space-borne broadband data (e.g. CERES) are very finite due to different orbiting patterns, further limiting the use of TROPOMI data in this study.

## 2.2 CERES Data

The Aqua Cloud and the Earth's Radiant Energy System (CERES) instrument measures upwelling radiant energy in
the shortwave (0.3 -5 µm), window (8-12 µm), and total spectra (0.3 – 100 µm) (Su et al., 2015a, b; Wielicki et al.,
1996). The spatial resolution for Aqua CERES is on the order of 20 km at nadir. In this study, we used upwelling top
of atmosphere (TOA) shortwave flux (SWF) data from the Aqua CERES Level 2 Single Scanner Footprint (SSF) data
product to assess the direct radiative effects of the biomass burning smoke in the Arctic. The CERES SSF data are
derived by colocating CERES observations with MODIS aerosol and cloud data to provide aerosol information and
for cloud screening of observed CERES scenes. The CERES SWF data are derived by converting from observed
radiances to fluxes using predetermined angular distribution models (ADMs), with different ADMs applied for
different surface types (land, snow type, sea ice, ocean) and cloud conditions (clear-sky, partly-cloudy, and cloudy)
(Su et al., 2015a, b).

CERES data have been used extensively for investigating changes in Arctic radiative energy budgets for both TOA
(Duncan et al., 2020; Kay and L'Ecuyer, 2013; Riihelä et al., 2013) as well as the surface (Boeke and Taylor, 2016;
Hegyi and Taylor, 2017). Previous studies have also worked to validate Arctic CERES surface radiative fluxes (Di
Biagio et al., 2021; Riihelä et al., 2017) and TOA fluxes (Taylor et al., 2022), with the latter seeking to validate CERES
TOA radiative fluxes against aircraft-based upwelling radiative flux observations. While Taylor et al. (2022) noted
some error in the Arctic CERES Level-2 SSF TOA upwelling SWF resulting largely from errors in the imager-based
SICs used in the scene classification, the CERES observations compared well overall with the aircraft observations
(differences between the CERES and aircraft observations were within $2\sigma$ uncertainty). The authors concluded that
CERES TOA radiative flux data are suitable for polar climate studies (Taylor et al., 2022).

## 2.3 MODIS Data

Along with the Aqua CERES data, we used multiple data products from the Aqua MODerate resolution Imaging
Spectroradiometer (MODIS), which measures spectral radiances in 36 channels across the visible, near-infrared, and
infrared spectra (Justice et al., 1998). Level 1B Aqua MODIS 2.1 µm reflectance (1-km spatial resolution, from data
product MYD021KM, (MODIS Characterization Support Team (MCST), 2017)) and Level 2 cloud optical depth and
cloud top pressure (1-km spatial resolution, from data product MYD06, (Platnick et al., 2015)) were used in this study
for identifying the visible thickness and height of clouds around the Arctic. Cloud detection in the Arctic is a
challenging problem, so we included the MODIS 2.1 µm reflectance for added confidence in cloud screening over
Arctic sea ice. Unlike clouds, which exhibit high reflectance from both the visible and 2.1 µm channels, sea ice and
snow look bright at the visible channel but have a low reflectivity at the 2.1 µm channel. Thus, reflectance data from
the 2.1 µm channel can be further used to assist cloud-clearing of CERES and OMI data over the Arctic region. For
example, for the Aqua MODIS granule over northeastern Russia and the Arctic Ocean, the OMI UVAI data (Fig. 2a)
and Aqua MODIS true color image (Fig. 2b) reveal dense smoke extending from northeastern Russia out over the
Arctic Ocean, as well as widespread bright features (cloud and ice) over the Arctic Ocean. However, from the visible

imagery, it is difficult to distinguish between clouds and sea ice, so we analyzed the MODIS 2.1 um reflectance data in the same regions (Fig. 2c). The 2.1 um reflectance values reveal a clear distinction between clouds and sea ice covered surfaces.

The MODIS 2.1 um reflectance data also help in minimizing the number of dense smoke plumes that are mistakenly classified as cloud. For example, in the dense biomass burning smoke plume over northeastern Russia shown in Fig. 2a and Fig. 2b, the L1B Aqua MODIS cloud mask (Fig. 2d) classifies about half of the plume as "cloudy" or "probably cloudy." However, the Aqua MODIS 2.1 μm reflectance (Fig. 2c) in the same plume region does not exhibit any higher values that would indicate the presence of cloud (for example, the higher 2.1 μm reflectance values across the

lower half and left third of the panel indicate clouds; note that this is reflected in the MODIS L1B cloud mask shown in Fig. 2d). The 2.1 μm reflectance in the plume region very closely matches the reflectance of the nearby clear regions, suggesting that there are no clouds within the dense smoke plume and the MODIS L1B cloud mask misclassified the dense smoke as cloud. The MODIS cloud mask also misclassified smoke over ice and ocean scenes as "cloud", as shown in Fig. 2d.


**Figure 2.** Comparison of (a) OMI UV aerosol index, (b) Aqua MODIS true color imagery, (c) Aqua MODIS 2.1 μm reflectance, and (d) Aqua MODIS L1B cloud mask overlaid on the MODIS 2.1 μm reflectance in a biomass burning smoke plume over northeastern Russia and the Arctic Ocean.

Level-3 daily-gridded Aqua MODIS cloud optical depth (subsets of the daily MYD08_D3 product, (Platnick, S. et al.,

2015a)) data were also used when calculating daily estimated Arctic ADRF. Level-3 monthly gridded Aqua MODIS cloud fraction (from the MYD08_M3 product (Platnick, S. et al., 2015b) from April through September of 2005 through 2020 were used for qualitative comparisons between Arctic-region cloud fraction trends and the observation-based ADRF trend estimates.

**2.4 SSMIS Sea Ice Concentration (SIC) Data**

The Defense Military Satellite Program (DMSP) Special Sensor Microwave Imager / Sounder (SSMIS) instruments are linearly polarized passive microwave radiometers that measure upwelling microwave radiances in 24 channels (Kunkee et al., 2008). The first SSMIS instrument was launched on board the DMSP F-16 spacecraft in 2003 (Kunkee et al., 2008). Version 2 daily sea ice concentration (SIC) data from DMSP SSMIS passive microwave data were obtained from the National Snow and Ice Data Center (NSIDC) data archive from April through September of 2005 through 2020 over the Arctic region on a 25 x 25 km polar stereographic grid (DiGirolamo et al., 2022). We used SSMIS daily sea ice data for determining surface types (ice, mixed ice/ocean, ocean, and land) in the Arctic region. We also used monthly SSMIS SIC data from the NSIDC data archive for qualitative comparisons between Arctic SIC trends and the ADRF trends.

The SSMIS SIC dataset used in this study is one of two key SIC datasets provided by the NSIDC and has been used extensively in the scientific community to study Arctic sea ice trends. The algorithm used in the dataset, developed by NASA (Cavalieri et al., 1984), has been included in several SIC validation studies (Cavalieri et al., 1992; Ivanova et al., 2015; Kern et al., 2019, 2020; Meier, 2005; Steffen and Schweiger, 1991). Overall, and as reported in the NSIDC dataset user guide (https://nsidc.org/sites/default/files/documents/user-guide/nsidc-0051-v002-userguide.pdf), errors in the SIC dataset are less than 5% in the wintertime but can be as large as 15% in the summertime (Cavalieri et al., 1992). Some recent studies have reported that the SIC dataset may underestimate SIC by up to 10% (Kern et al., 2019, 2020), with the underestimation being partly caused by surface melt ponds in the summer months (Steffen and Schweiger, 1991). Additionally, microwave-based sea ice concentrations have been found to be sensitive to areas of thin ice (Ivanova et al., 2015). Nevertheless, despite some limitations, the algorithm and associated SIC dataset are widely used to represent Arctic SIC.

**3. Estimate instantaneous ADRF over multiple surface types and cloud conditions**

Aerosol radiative forcing is defined relative to the aerosol-free conditions, given by:

$$ADRF = SWF_{cln} - SWF_{all} \qquad (2)$$

where $SWF_{cln}$ is the aerosol-free SWF, $SWF_{all}$ is the all-sky SWF, and $ADRF$ is the aerosol direct radiative forcing. With recent work exhibiting the utility of OMI UVAI data at identifying BB aerosol plumes over the bright Arctic ice and cloud surfaces (Sorenson et al., 2023), UVAI data can serve as the basis for quantifying instantaneous absorbing aerosol radiative forcing in the Arctic region with co-located satellite observations. While the vertical aerosol distribution significantly affects the retrieved UVAI values, we do not have the proper observations of aerosol vertical distribution to accurately account for these effects in this analysis, so the aerosol layer height was not included as a variable in estimating ADRF over the Arctic region. This is also partially because the impact of aerosol extinction profiles has less effect on clear sky TOA ADRF (Guan et al., 2010).

To estimate ADRF, all swaths of OMI UVAI data were scanned to identify OMI swaths that contained widespread and significant (perturbed UVAI > 2.0) absorbing aerosol events over the Arctic region, which we defined here as north of 65 °N. We chose the 2.0 UVAI threshold for this step to efficiently select swaths that provide good coverage of high aerosol loading conditions, with those swaths also containing regions of lower UVAI to provide good data range for machine learning training purposes. We identified 131 OMI swaths meeting these criteria, and for each of these swaths, each OMI pixel north of 65 °N was co-located with a L2 Aqua CERES TOA SWF and surface albedo, a L1B Aqua MODIS 2.1 μm reflectance and L2 Aqua MODIS cloud optical depth and cloud top pressure, and a 25 x 25 km SSMIS SIC value. As described in Section 2.1, we used the L2 OMI quality control flags and the methods described by Sorenson et al. (2023) to exclude pixels with flagged or unflagged OMI row anomaly contamination from the co-located dataset. Due to the similar pixel size between the OMI footprint (13 x 24 $km^2$ near nadir) and both the CERES footprint (20 x 20 $km^2$ near nadir) and the SSMIS grid box (25 x 25 $km^2$), we applied a "nearest-neighbor" approach to co-locate the nearest CERES pixel and SSMIS grid box to each OMI pixel. We excluded pixels from the co-located dataset with the SSMIS surface type flag denoting coastline pixels or pixels too close to the North Pole (i.e. in the "pole hole"). However, additional averaging steps were required for colocating the MODIS data to the OMI grid because the MODIS pixels (1 x 1 $km^2$) are much smaller than the OMI pixels. For all MODIS products (2.1 μm reflectance, cloud optical depth, and cloud top pressure), the co-location values consisted of the averages of the values from all MODIS pixels with latitudes and longitudes that were within the latitude/longitude bounds of the OMI pixel, with these bounds defining the latitudes and longitudes of the four corners of each OMI pixel (provided with the OMI data). For the MODIS cloud top pressure data, an additional check was added to ensure only non-zero cloud top pressure values were included in the averaging for each co-location pixel. This was done to avoid the skewing of the average cloud top pressure by some non-retrieval fill values of 0. After performing the co-location, each swath contains the following variables listed in Tab. 1.

**Table 1.** The variables contained in each co-located Arctic OMI L2 swath. The methods by which the CERES, MODIS, and SSMIS data are co-located to the L2 OMI grid, as well as the quality control methods applied to all the data products, are given in the right column.

| Sensor | Variable | Co-location/QA Processes Applied |
|---|---|---|
| OMI | UVAI Perturbation | Viewing geometry-based and surface type-based uncertainties removed following the "perturbing method" of Sorenson et al (2023). The row anomaly quality control flag was also applied to exclude rows impacted by the OMI row anomaly. |
| | Solar Zenith Angle (SZA) | Taken from L2 OMI data |
| | Viewing Zenith Angle (VZA) | Taken from L2 OMI data |
| | Latitude | Taken from L2 OMI data |
| | Longitude | Taken from L2 OMI data |
| Aqua CERES | TOA upwelling shortwave flux (SWF) | CERES value from nearest grid point |
| | Surface Albedo | CERES value from nearest grid point |

| Aqua MODIS | 2.1 μm reflectance | Pixels within the latitude and longitude bounds of each OMI pixel averaged together |
| --- | --- | --- |
| | Cloud Optical Depth (COD) | Pixels within the latitude and longitude bounds of each OMI pixel averaged together |
| | Cloud Top Pressure (CTP) | Pixels within the latitude and longitude bounds of each OMI pixel averaged together. Excluded values with CTP equal to 0. |
| SSMIS | Sea Ice Concentration (SIC) | SSMIS value from the nearest grid point |
| | | Coastline (sea ice value = 253) and "pole hole" (sea ice value = 251) removed |

Figure 3 shows the distribution of surface types identified for the absorbing aerosol-containing OMI pixels (plotted here for UVAI greater than 1.0) from each of the identified swaths from 2005 – 2020. The stacked bars represent the percent of aerosol-containing OMI pixels in the swath over the Arctic that fall into each surface category, with blue
corresponding to "ice" (SSMIS SIC greater than 80%), orange corresponding to "mix" (SSMIS SIC greater than 20% and less than 80%), green corresponding to "ocean" (SSMIS SIC less than 20%), red corresponding to "land" (SSMIS grid box containing a land mask value), and purple corresponding to "other" (SSMIS grid box identified as coastline or "pole hole"). Coastline pixels were excluded from this analysis to ensure only pixels that are entirely "land" were classified as such in our analysis. Land, ocean, and mixed ice/ocean conditions were frequently observed among the
swaths, with a smaller percent coverage of ice conditions observed. A table describing each of the smoke plumes being analyzed here, including their source region and visual characteristics, is included as an appendix. The majority of the identified swaths are from the boreal summer months (June, July, and August), times of frequent biomass burning events in northern Russia and Canada.

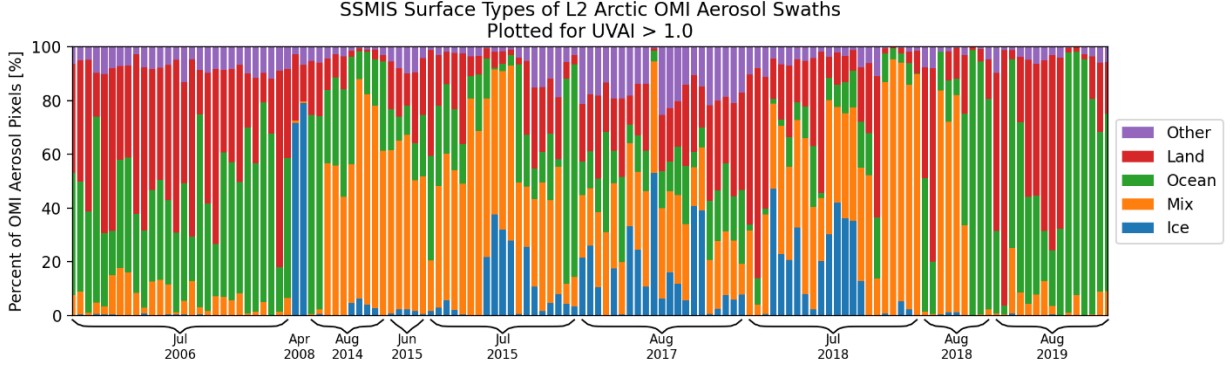

**Figure 3.** Distribution of SSMIS-derived surface types among the OMI pixels with UVAI > 1.0 in each of the selected aerosol-containing swath between April through September of 2005 through 2020.

After colocating the data, we inspected each of the selected aerosol swaths to determine if the locations of the BB aerosol plumes shifted between the MODIS imagery and OMI UVAI. Though Aqua and Aura are both members of the A-train satellite constellation, Aqua crosses the equator about 15 minutes before Aura, and thus likewise OMI

UVAI observations lag behind the Aqua MODIS and CERES observations by about 15 minutes. This time lag could introduce a shift in the BB plume locations in the OMI and MODIS data. Feng and Christopher (2015), who conducted a study of above-cloud aerosol radiative effect (ARE) over marine stratocumulus clouds near equatorial Africa using co-located observations from sensors on-board Aqua, Aura, and CALIPSO, assumed that the locations of the BB aerosol plumes studied did not significantly shift between the overpasses of the three satellites. Nevertheless, we

compared the locations of the plume identified in the OMI UVAI data with MODIS true color imagery for each aerosol swath, and the comparison of OMI and MODIS plume locations for the 03:08 UTC 10 August 2019 OMI swath is shown in Fig. 4. The OMI UVAI data are overlaid on the MODIS true color imagery in Fig. 4c, and the region of high UVAI lines up very closely with the visibly dense smoke in the true color imagery, showing that there was no significant drift between the OMI and MODIS observations for this plume. After visually inspecting all OMI and

MODIS plume location comparisons, we did not find any significant drift in any of the analyzed aerosol swaths.

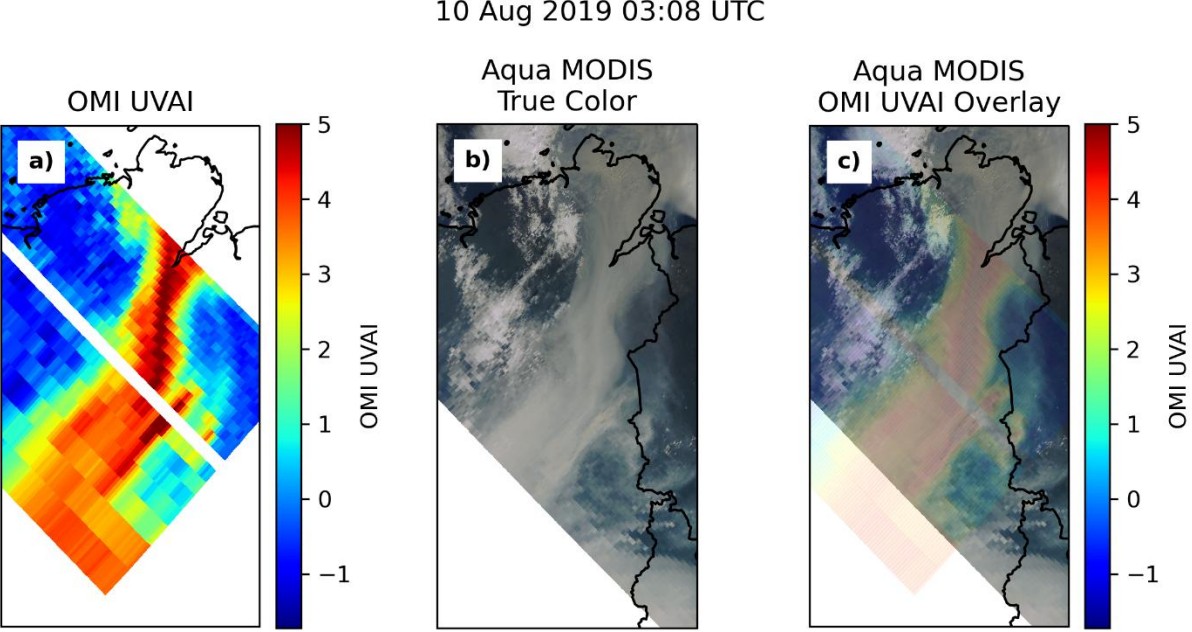

**Figure 4.** Comparison of BB smoke aerosol plume locations as observed by Aqua MODIS and OMI for the 10 August 2019 03:08 UTC OMI swath. a) OMI UVAI perturbations. b) Aqua MODIS true color imagery. c) Aqua MODIS true color imagery with the 300 OMI UVAI data overlaid.

### 3.1 Neural network for estimating aerosol-free SWF from L2 satellite data

With aerosol forcing being defined relative to the aerosol-free conditions, the difficulty of determining aerosol forcing from only observations is the determination of the aerosol-free TOA upwelling SWF in smoky regions. To solve this issue, we constructed and trained a neural network (NN) to estimate aerosol-free TOA upwelling SWF from

observations in BB smoke plumes around the Arctic. The NN was designed to take inputs of solar zenith angle; viewing zenith angle; SSMIS SIC and surface type; MODIS cloud optical depth, cloud top pressure, and 2.1 μm

reflectance; and CERES surface albedo, values that were assumed to be largely independent of the aerosols, and return aerosol-free SWF after training the NN on aerosol-free input data. We note that the assumption of the NN input variables being independent of the aerosol loading may not hold for the MODIS cloud optical depth data, as it is well

known that, due to the aerosol indirect effect, aerosol particles can greatly impact cloud properties such as cloud optical depth. Nevertheless, for simplicity, we designed this system to focus on the aerosol direct effect and leave the study of the impacts of the aerosol indirect effect on these results to a future study.

First, to provide a large training and testing dataset, we retrieved additional L2 OMI, MODIS, SSMIS, and CERES data from the four days on either side of each identified aerosol event, with the four-day window providing coverage

of the aerosol-free conditions in the aerosol regions. For example, for the smoke event of 24 – 27 July 2006, additional data were downloaded to fill in the time period of 20 – 31 July 2006. The other L2 swaths from the days with chosen aerosol-containing swaths were also included in the training dataset. Thus, a total of 116 days (each day may contain multiple OMI swaths) of L2 OMI, MODIS, SSMIS, and CERES observations were downloaded and co-located for training and testing purposes. However, since the NN needs to estimate aerosol-free SWF, and to ensure the validity

of the results when applying the NN to the aerosol swaths, the 131 aerosol-containing swaths were removed from the input dataset. Additionally, we removed 50 other randomly-selected swaths from the dataset and reserved them for validation of the NN model after training. Thus, about 1100 L2 OMI swaths with co-located MODIS, SSMIS, and CERES data were available for the NN training and testing dataset. As an extra check, all remaining pixels with OMI UVAI greater than 1 were removed to further ensure that only aerosol-free data were provided to the NN for training.

In addition to the OMI UVAI check, pixels were removed if the latitude was less than 65° N, the SSMIS SIC contained coastline or "pole hole" values, or the COD data were greater than 70. Since the Arctic SSMIS data containing land pixels are set to 254, these values were changed to 101 to remove the large discontinuity of the SIC data from 100 to 254. After applying all of these quality control checks and preprocessing steps, all of the values in each variable were scaled to a 0 – 100 range, ensuring all variables were equally weighted in the NN (thus, the min and max of each

variable sent to the NN was 0 and 100, respectively). The CERES SWF values used for validating the model at each training epoch were also scaled to a 0 to 100 range. Following these QC steps, there were 4.4 million available aerosol-free pixels identified, and with 10% of these being reserved for testing purposes (about 400,000 pixels), the training dataset consisted of 4 million aerosol-free pixels.

Figure 5 shows the architecture of the NN, which consists of 13 total layers: one input layer, 11 fully-connected hidden

layers, and 1 output layer. The input layer consists of 7 nodes, with one node for each input variable (solar zenith angle, viewing zenith angle, SIC / surface type, MODIS 2.1 μm reflectance, MODIS cloud optical depth, MODIS cloud top pressure, and CERES surface albedo). The hidden layers contain an increasing number of nodes per layer from hidden layer 1 to layer 6, with layers 1 through 6 having 8, 12, 16, 24, 32, and 64 nodes, respectively, and a decreasing number of nodes per layer from layers 6 to 11, with layers 7 through 11 having 32, 24, 16, 12, and 8 nodes,

respectively. Finally, the output layer consists of 1 node that uses linear activation. With all of the input variables being scaled to a 0 – 100 scale, the output SWF value is on a 0 – 100 scale, so this output value was reverted to a true

SWF using the same scaling values to convert the original SWF values to the 0 – 100 scale. The NN was built using the TensorFlow Python package (Abadi et al., 2015).

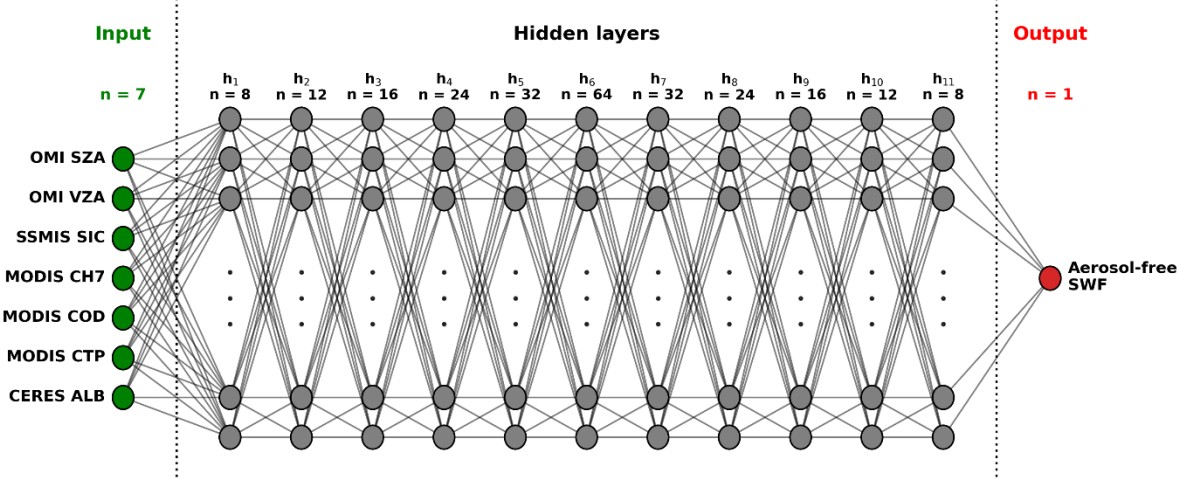

**Figure 5**. Architecture of the neural network for estimating L2 aerosol-free SWF from L2 input values of solar zenith angle (SZA), viewing zenith angle (VZA), sea ice concentration (SIC), 2.1 μm reflectance (CH7), cloud optical depth (COD), cloud top pressure (CTP), and surface albedo (ALB). Green circles represent nodes in the input layer, gray circles represent nodes in the hidden layers, and the red circle represents the node in the output layer. All nodes in the neural network are fully connected to the nodes in the next layer, as illustrated by the lines connecting the circles.

Several experiments were conducted to determine the best activation function (AF) to use in the NN hidden layers. The NN was trained multiple times using different AFs in the hidden layer nodes, and the ending mean absolute errors (MAE) of the NN-predicted aerosol-free SWF against CERES SWF observations after training with each AF for 100 epochs are listed in Tab. 2. The Leaky Rectified Linear Unit (LeakyReLU, Maas et al., 2013) AF gave the best performance with an ending MAE of 2.86 Wm$^{-2}$, while the Rectified Linear Unit (ReLU, Nair and Hinton, 2010) AF gave the second-best performance with an ending MAE of 2.92 Wm$^{-2}$. With the LeakyReLU activation function known to avoid the "dead neuron" problem associated with the ReLU activation function (Dubey et al., 2022; Maas et al., 2013), we suspect that this could be behind the slightly better performance of the LeakyReLU AF relative to the ReLU AF. Other models that gave good performance, but slightly worse performance than LeakyReLU, include the softplus (Glorot et al., 2011) and softsign (Glorot and Bengio, 2010) AFs, though the simulation with the softplus AF exhibited some instability between epochs 60 and 80. While the experiments with Exponential Linear Unit (ELU, Clevert et al., 2016) and Scaled Exponential Linear Unit (SELU, Klambauer et al., 2017) AFs ended with MAE of around 3.2 Wm$^{-2}$, the training was highly unstable, with the errors spiking randomly between 3.0 Wm$^{-2}$ and 3.5 Wm$^{-2}$ with each epoch. The linear AF provided one of the worst performances with an ending MAE of 5.47 Wm$^{-2}$, while the training experiments with Gaussian Error Linear Unit (GELU, Hendrycks and Gimpel, 2016) and sigmoid AFs were stopped early because the MAE after the first about 10 epochs remained at around 12 Wm$^{-2}$ and did not converge. Since the LeakyReLU activation function gave the best performance out of the other activation functions tested in this experiment, we used this activation function in all NN hidden layer nodes during training. Training was conducted on a GPU node for 100 epochs with a batch size of 128, an Adam optimizer (Kingma and Ba, 2017), and with back-

propagational loss being derived by minimizing the mean squared error. After training for 100 epochs, the mean squared error (MSE) and mean absolute error (MAE) of the model-estimated SWF values against the training observations were 16.9 Wm$^{-2}$ and 2.86 Wm$^{-2}$, respectively.

**Table 2. Mean absolute errors (MAE) of the neural network output after training for 100 epochs with several different activation functions. Training with the sigmoid and GELU activation functions was terminated after about 10 epochs due to the extremely high MAE and the lack of convergence during the training process.**

| Activation Function | Reference | Mean absolute error after training for 100 epochs (Wm$^{-2}$) |
|---|---|---|
| LeakyReLU | (Maas et al., 2013) | 2.86 |
| ReLU | (Nair and Hinton, 2010) | 2.92 |
| Softplus | (Glorot et al., 2011) | 2.94 |
| Softsign | (Glorot and Bengio, 2010) | 3.06 |
| ELU | (Clevert et al., 2016) | 3.21 |
| SELU | (Klambauer et al., 2017) | 3.32 |
| Tanh | | 4.87 |
| Linear | | 5.47 |
| Sigmoid | | ~12* |
| GELU | (Hendrycks and Gimpel, 2016) | ~12* |

## 3.2 Validation of the NN against CERES

Once trained, the NN was first applied to the 50 reserved aerosol-free validation swaths (independent from the 131 aerosol swaths) to validate the NN output against CERES observations. The 50 validation swaths contained about 200,000 pixels to use for validation; we note that similar validation results were obtained when increasing the size of the validation dataset to about 300,000 pixels by adding 25 additional aerosol-free OMI swaths (and co-located MODIS, SSMIS, and CERES data) randomly chosen from the 2005 – 2020 boreal summer study period. Errors were calculated between the NN-estimated aerosol-free SWF and the associated CERES TOA SWF observations, and the distribution of the errors from the 50 validation swaths is shown in Fig. 6a. The error distribution peaks at about 0 Wm$^{-2}$, suggesting little overall bias in the NN-estimated aerosol-free SWF values. To further test for systematic biases in the NN-estimated aerosol-free SWF, we binned the validation dataset first by the SSMIS SIC and surface type, and then by MODIS COD. The NN error distributions binned by the SSMIS surface type and the MODIS COD are shown in Fig. 6b and Fig. 6c, respectively. We found that the mean SWF errors for the error distributions binned by SSMIS SIC and MODIS COD are largely small, with magnitudes primarily less than 3 Wm$^{-2}$. The peaks of nearly all the error distributions for the different surface types and CODs are around 0 Wm$^{-2}$, suggesting little systematic bias in the system associated with the different surface types and CODs. The mean error for the land distribution (Fig. 6b, brown) is slightly larger at -5.5 Wm$^{-2}$, suggesting a slight negative bias over land. We suspect that this is related to the lack of information about the land-based surface type in the system. If the NN is primarily trained on dark land surfaces, but it is applied to brighter-than-normal land surfaces (e.g. snow- and ice-covered land), the NN will predict lower

upwelling SWF than is observed by CERES. When excluding data from April and May from this analysis, the mean error for the over-land data is much smaller, supporting our hypothesis that the slight negative shift in the land-based error distribution is related to the land surface brightness that is unaccounted for in this system. Given that the majority of the smoke events analyzed in the study occurred in the summer months (June – August), we do not expect this potential low bias of the NN over bright land surfaces to significantly impact the results of our study.

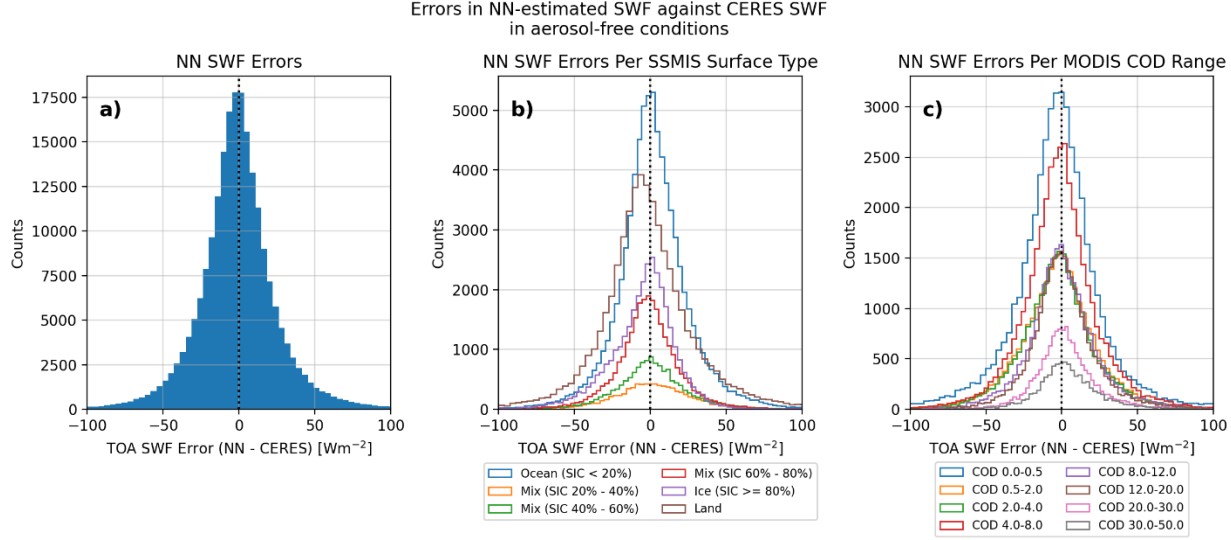

**Figure 6.** a) Distribution of errors in the neural network (NN)-estimated aerosol-free shortwave flux (SWF) relative to CERES TOA upwelling SWF observations for the 50 validation swaths reserved from the NN training dataset. b) As in (a), but with the errors binned by the SSMIS sea ice concentration (SIC) and surface type. c) As in (a), but with the errors binned by MODIS cloud optical depth (COD).

### 3.3 Analysis of NN-based ADRF estimates on L2 basis

We then applied the NN to the 131 aerosol swaths to estimate aerosol-free SWF in smoky regions. Comparisons of the NN-estimated aerosol-free SWF against OMI UVAI and CERES SWF observations for an aerosol-free swath (22:44 UTC 8 July 2018, one of the 50 swaths reserved for validation) and a swath containing an aerosol plume (22:13 UTC 5 July 2018, one of the 131 aerosol swaths) are shown in Fig. 7. The OMI UVAI perturbations for the first swath (Fig. 7a) are all below 0, confirming that there were no absorbing aerosols within the swath. The aerosol-free SWF values generated by the neural network (Fig. 7c) closely match the observed CERES SWF values (Fig. 7b), with the NN-estimated SWF matching both the patterns and intensity of the observations. Fig. 7d, which shows the spatial differences between the CERES-observed SWF against the NN-estimated SWF, shows small differences between the observations and NN output; the $R^2$ of the comparison between the CERES observations and NN output (shown in Fig. 7i) is 0.955 showing the overall agreement between the observations and NN output. For the aerosol swath, the OMI UVAI observations (Fig. 7e) exhibit a plume of high UVAI (> 4) perturbations extending from far northeastern Russia out over the Arctic Ocean, over regions of both sea ice and open ocean water. The CERES SWF observations (Fig. 7f) and NN-estimated aerosol-free SWF values (Fig. 7g) show agreement across much of the swath, in regions with very low OMI UVAI, but the differences between the CERES and NN SWF values (Fig. 7h) reveal large

differences within the plume region. In the plume areas over dark land surfaces and ocean water, the NN – CERES differences are strongly negative, showing that the NN output is much lower than the CERES observations within

420    these regions; this agrees with the expected behavior of a dense smoke plume over a dark surface, with the BB smoke scattering sunlight upwards to TOA and inducing a strong cooling effect from TOA. On the other hand, the differences between the CERES and NN SWF values in the plume regions over cloudy and icy regions are strongly positive, showing that the NN-estimated aerosol-free SWF values are much higher than the CERES observations; this suggests that the BB aerosols have a darkening (or warming) effect over the bright cloud and ice surfaces. We note that some

425    regions in the NN-estimated aerosol-free SWF values contain missing values, which we suspect is a result of missing L2 MODIS COD values. Even in the aerosol-containing swath, after removing pixels with OMI UVAI perturbation greater than 1, the $R^2$ of the comparison between the CERES observations and NN output (shown in Fig. 7j) is still high at 0.933. Using the CERES observations and NN output shown in Fig. 7i and j, we calculated a noise floor of about 18 Wm$^{-2}$.

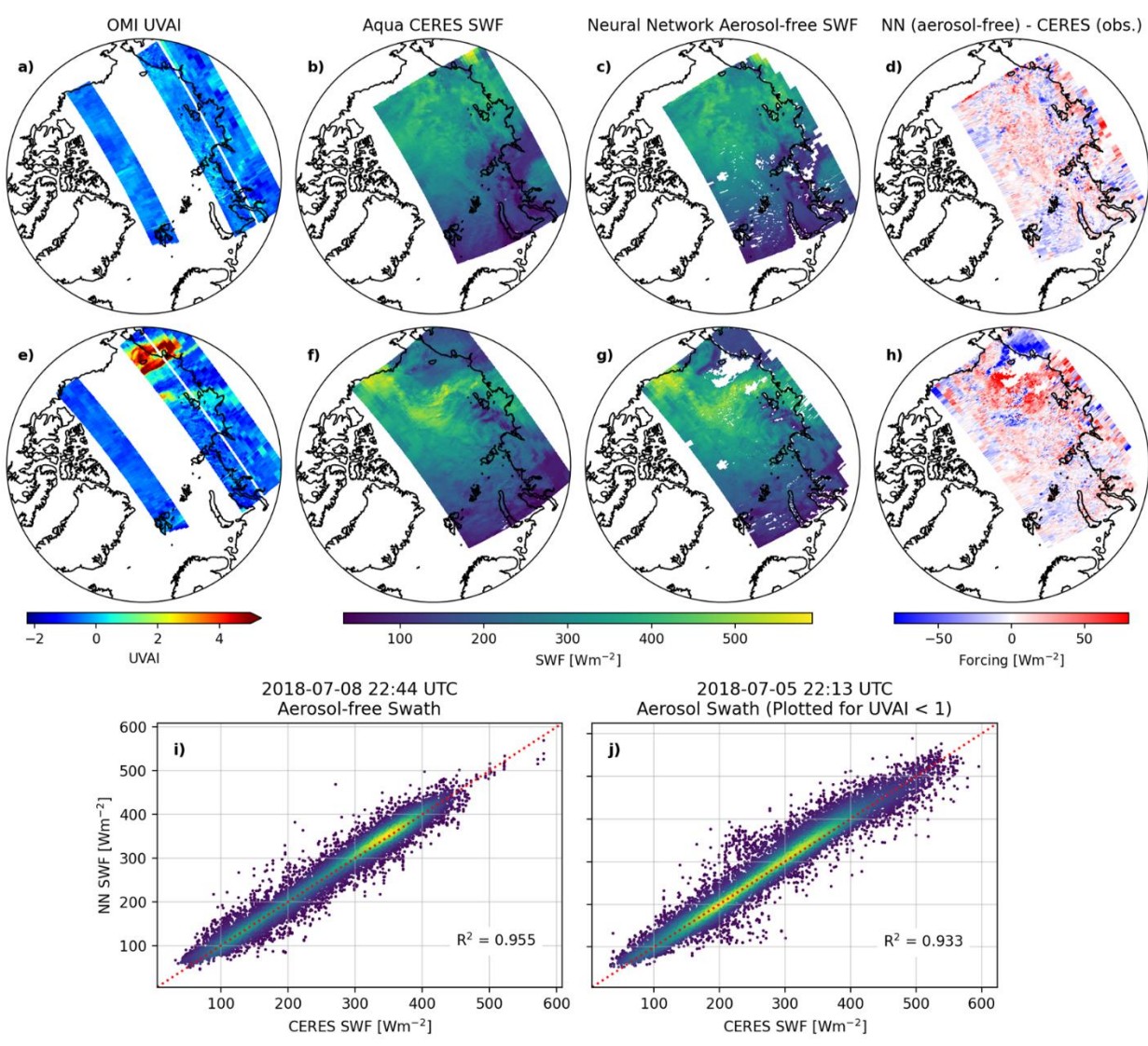

**Figure 7.** Validation of the neural network under aerosol-free (top row) and aerosol (middle row) conditions. The first two rows contain maps of the (first column) OMI UVAI perturbations, (second column) CERES SWF observations co-located to the OMI grid, (third column) NN-estimated aerosol-free SWF, and (fourth column) difference between the NN-estimated aerosol-free SWF and CERES SWF observations. The bottom row contains scatter plots of the CERES SWF and NN aerosol-free SWF from the (i) aerosol-free swath and (j) aerosol-containing swath, with points plotted for the aerosol-containing swath only when the OMI UVAI was less than 1.

## 4. Estimate long-term trends in observation-based ADRF

### 4.1 Generate a look-up table (LUT) of aerosol forcing regression statistics from binned L2 data

While aerosol direct forcing trend can be directly estimated using CERES data and neural network simulated aerosol free TOA SWF as mentioned in Section 3, it is rather computationally expensive to perform those estimations on 16 years of Level 2 data. As an alternative, ADRF values can be estimated at the OMI UVAI domain. In this approach,

aerosol direct forcing values from Section 3 were used to derive the relationship between ADRF and observing conditions, including the underlying surface conditions (e.g. sea ice, clouds, oceans, land), aerosol loading (proxied by OMI UVAI) and observing angles (e.g. SZA, VZA). Upon validating against aerosol forcing values using approaches as mentioned in Section 3, long-term aerosol forcing trend (at the OMI UVAI domain) and uncertainties were derived using an innovative, Monte-Carlo-based method, and through the analysis of daily level 3 (L3) cloud, sea ice and OMI UVAI data.

In this approach, aerosol forcing efficiency, which is defined as ADRF per OMI UVAI in this study, was estimated based on observing conditions including OMI solar zenith angle (SZA), MODIS cloud optical depth, and SSMIS SIC. The observation conditions were quantified in discrete size bins, and we used uniform bin sizes of 5° and 20% for the OMI SZA and SSMIS SIC, respectively, with MODIS COD bin sizes increasing from 0.5 for low COD values to 20 for high COD values. Examples of deriving aerosol forcing efficiency as functions of observing condition are shown in Fig. 8 for each of the SSMIS surface type categories and COD bins. The data for each of the surface type bins in this figure are not separated by solar zenith angle, but as shown later in Fig. 9, the ADRF does not change significantly with solar zenith angle. A table containing the mean and standard deviation of the ADRF for three of the COD bins is given in Tab. 3. The magnitudes and signs of the ADRF vary significantly as a function of COD and SIC.

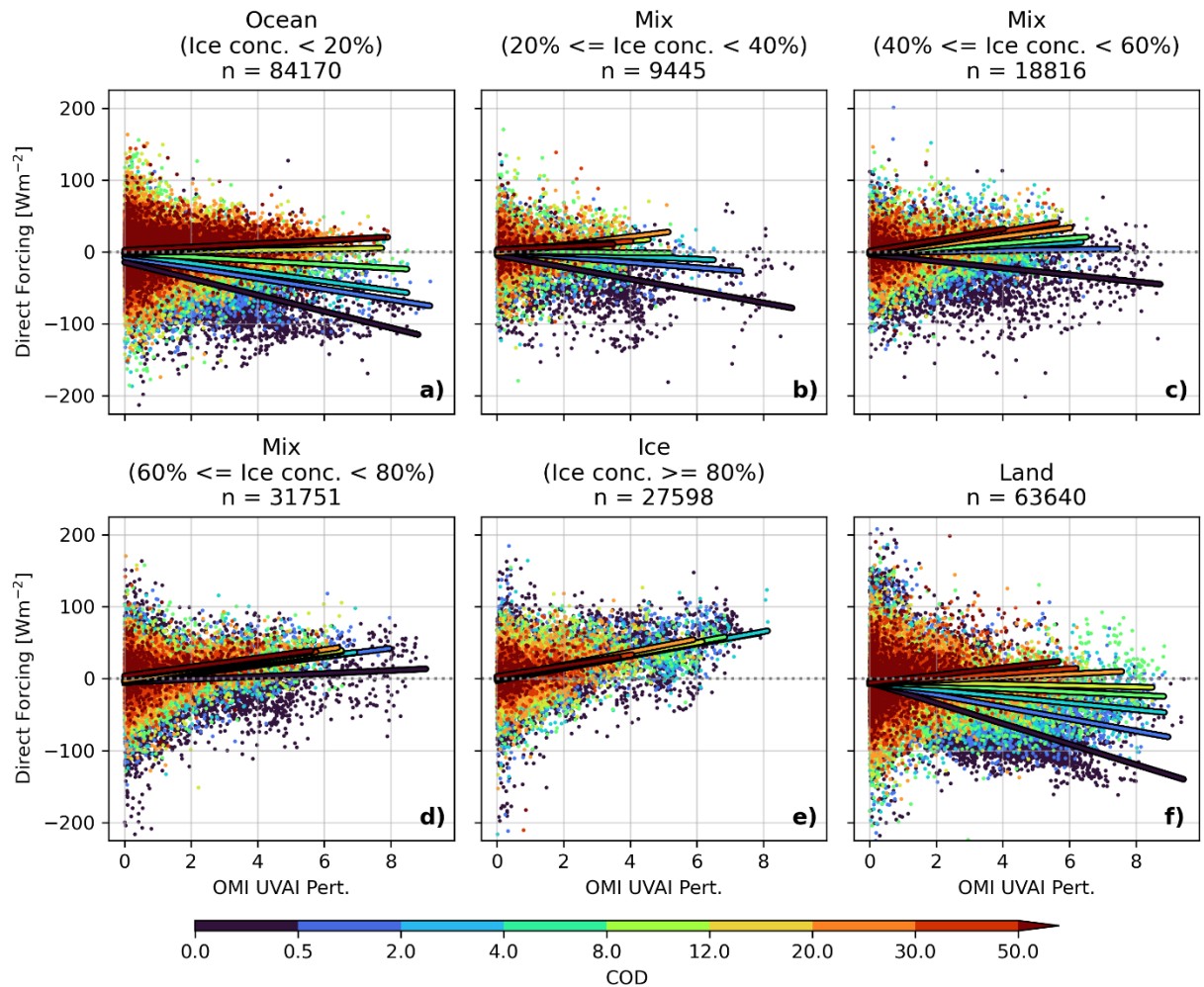

**Figure 8.** Aerosol direct radiative forcing (ADRF) derived from co-located satellite observations (CERES, OMI, MODIS, SSMIS) and neural network output of aerosol-free SWF, divided by COD and binned for ocean surfaces (sea ice concentration (SIC) below 20%, panel a), mixed ice/ocean surfaces (SIC between 20% and 40%, panel b; SIC between 40% and 60%, panel c; SIC between 60% and 80%, panel d), ice surfaces (SIC greater than 80%, panel e), and land surfaces (panel f). Linear regression lines between ADRF and OMI UVAI are plotted for each of the COD bins. Counts of L2 pixels in each surface type bin are given in the subplot titles.

**Table 3.** Mean and standard deviation (in Wm$^{-2}$) of the absorbing aerosol direct radiative forcing (ADRF) from Fig. 8 binned by SSMIS sea ice concentration (SIC) / surface type, MODIS cloud optical depth (COD), and OMI UV aerosol index (UVAI). Results are given for three COD ranges: 0 – 0.5, 8 – 12, and 20 – 30.

| Mean and Standard Deviation of Binned L2 ADRF | | | | | | | | | |
|---|---|---|---|---|---|---|---|---|---|
| **SSMIS Surface Type** | **MODIS COD** | **UVAI 0 – 2** | | **UVAI 2 – 4** | | **UVAI 4 – 6** | | **UVAI > 6** | |
| | | **Mean ADRF** | **ADRF St.Dev.** | **Mean ADRF** | **ADRF St.Dev.** | **Mean ADRF** | **ADRF St.Dev.** | **Mean ADRF** | **ADRF St.Dev.** |
| **Ocean (0% - 20% ice)** | 0 – 0.5 | -21.7 | 28.3 | -47.8 | 32.0 | -66.4 | 38.9 | -78.4 | 32.9 |
| | 8.0 – 12 | -1.0 | 24.8 | 1.5 | 19.8 | 3.3 | 14.6 | 2.6 | 15.8 |
| | 20 – 30 | 0.6 | 20.7 | 8.4 | 14.5 | 10.4 | 18.2 | 14.2 | 11.8 |

| | | | | | | | | |
|---|---|---|---|---|---|---|---|---|
| **Mix (20% - 40% ice)** | **0 – 0.5** | -8.8 | 31.8 | -32.0 | 35.1 | -51.7 | 37.6 | -35.2 | 39.7 |
| | **8.0 – 12** | 1.7 | 29.0 | 13.1 | 22.8 | 33.7 | 29.7 | N/A | N/A |
| | **20 - 30** | 4.4 | 24.9 | 12.9 | 31.6 | 12.3 | 0.0 | N/A | N/A |
| **Mix (40% - 60% ice)** | **0 – 0.5** | -6.3 | 30.3 | -15.6 | 32.8 | -32.1 | 38.6 | -27.7 | 33.8 |
| | **8.0 – 12** | 1.8 | 25.8 | 14.4 | 22.8 | 31.8 | 19.1 | 42.9 | 0 |
| | **20 - 30** | 3.7 | 22.3 | 18.1 | 16.9 | 43.3 | 26.8 | N/A | N/A |
| **Mix (60% - 80% ice)** | **0 – 0.5** | -4.0 | 32.7 | 1.3 | 25.7 | 1.3 | 33.3 | 16.3 | 34.1 |
| | **8.0 – 12** | 2.1 | 25.5 | 16.6 | 22.0 | 26.9 | 23.7 | 83.3 | 14.9 |
| | **20 - 30** | 5.8 | 23.3 | 21.0 | 17.2 | 32.2 | 18.9 | N/A | N/A |
| **Mix (0% - 20% ice)** | **0 – 0.5** | 4.5 | 27.6 | 26.1 | 29.9 | 36.5 | 32.0 | 62.3 | 25.9 |
| | **8.0 – 12** | 5.7 | 25.1 | 25.0 | 27.4 | 43.9 | 27.6 | 44.6 | 15.1 |
| | **20 - 30** | 7.5 | 22.5 | 23.3 | 20.6 | 24.6 | 0.0 | N/A | N/A |
| **Land** | **0 – 0.5** | -14.5 | 34.1 | -50.2 | 32.8 | -75.5 | 31.8 | -71.1 | 40.7 |
| | **8.0 – 12** | -3.7 | 35.2 | -6.3 | 26.1 | -13.8 | 29.5 | -12.7 | 28.9 |
| | **20 - 30** | -4.9 | 32.4 | 5.8 | 28.3 | 5.0 | 24.3 | 25.8 | 17.1 |

For primarily cloud-free scenes (COD < 0.5, dark blue in Fig. 8), ADRF over dark surfaces such as ice-free ocean and land is strongly negative (i.e., scene brightened). For high UVAI scenarios and for COD < 0.5, the ADRF for both land and ocean conditions is as large as -100 Wm$^{-2}$, indicating a strong TOA cooling effect of dense aerosol plumes. For the same low COD conditions, the forcing for high UVAI scenarios increases gradually with increasing SSMIS SIC, with the sign of the forcing efficiency switching from negative to positive between the 40% – 60% and 60% – 80% bins, or roughly a SIC of 60% (we note that a similar threshold of 60% - 65% is also found when binning the ADRF data using a variety of other SIC bin sizes and bin edges). However, for primarily cloud-free scenes over sea ice (SSMIS SIC >= 80%), forcing over the bright surfaces is strongly positive, with ADRF values for high UVAI scenarios being as large as +80 Wm$^{-2}$ (i.e., scene darkening). As COD increases, the ADRF as a function of UVAI also generally increases (darkening), though the increase per unit COD is higher for darker surfaces than for lighter surfaces. The change in forcing efficiency (the slope of the UVAI vs ADRF regression line) as COD increases is large for ocean and land surfaces, but the slopes of the lines remain roughly the same over ice scenes. The slopes of the UVAI vs ADRF regression lines are positive across all surface types for high COD (> 20) scenes, suggesting that the thick clouds obscure the ocean and land surfaces below. The regression equations for the data plotted in Fig. 8 are listed in the appendix (Tab. A2).

Following similar steps, for each OMI SZA, MODIS COD, and SSMIS surface type bin, all ADRF values and associated OMI UVAI values were analyzed with linear regression to identify the slope and intercepts of the fitted line between the data (we note that similar results were found when using the more robust Theil-Sen slope estimator). The slopes and intercepts of the forcing regression allowed the ADRF to be estimated given an input UVAI value and the associated SZA, COD, and sea ice value, and these values were used to construct a look-up table (LUT) of aerosol forcing regression statistics for use in estimating daily aerosol direct forcing from L3 data. The slopes of the regressions applied to each bin for each surface type are shown in Fig. 9. Only grid boxes with more than 50 co-located values in the box are shown in the figure. Over ocean and land surfaces, negative forcing efficiencies were

identified for low COD conditions, shown by the blue on the bottom of these panels. The negative forcing efficiency (negative slope) shows that an increase in UVAI leads to negative ADRF, meaning the presence of the aerosols leads to increased upwelling TOA SWF; in other words, negative forcing leads to less energy into the Earth and Atmospheric

system (a brightening effect). The negative forcing efficiencies for low COD conditions over land suggest that the land surfaces were dark in the input data, which is not surprising given that most of the input data for the NN were from boreal summer. Thus, we did not detect data data with land snow coverage in this analysis. For COD values primarily above 8 over land and ocean scenes, the magnitude of the forcing efficiency slopes shift to being positive; as a note, this behavior closely matches results reported by Feng and Christopher (2015) in their analysis of aerosol

forcing over tropical marine stratocumulus clouds. The positive forcing efficiencies for the higher COD values indicates that higher aerosol loading leads to less upward-directed SWF (i.e., a darkening effect). On the other hand, over ice surfaces, the forcing efficiencies are entirely positive, with little change exhibited for increasing COD over icy surfaces. Over mixed ice/ocean surfaces, there is some variability with increasing COD, with slightly negative forcing efficiency for clear-sky conditions (COD < 0.5) and positive forcing efficiency for nearly all other SZA and

COD bins. While not shown, the slope standard errors in most of the bins are generally low (< 2 $Wm^{-2}$ $UVAI^{-1}$), though higher slope errors are found in some of the outer bins where the number of co-located values per bin is very low, such as for the "Mix 20% - 40%" bin. These results show the complex nature of aerosol forcing over the different cloud and surface conditions in the Arctic.

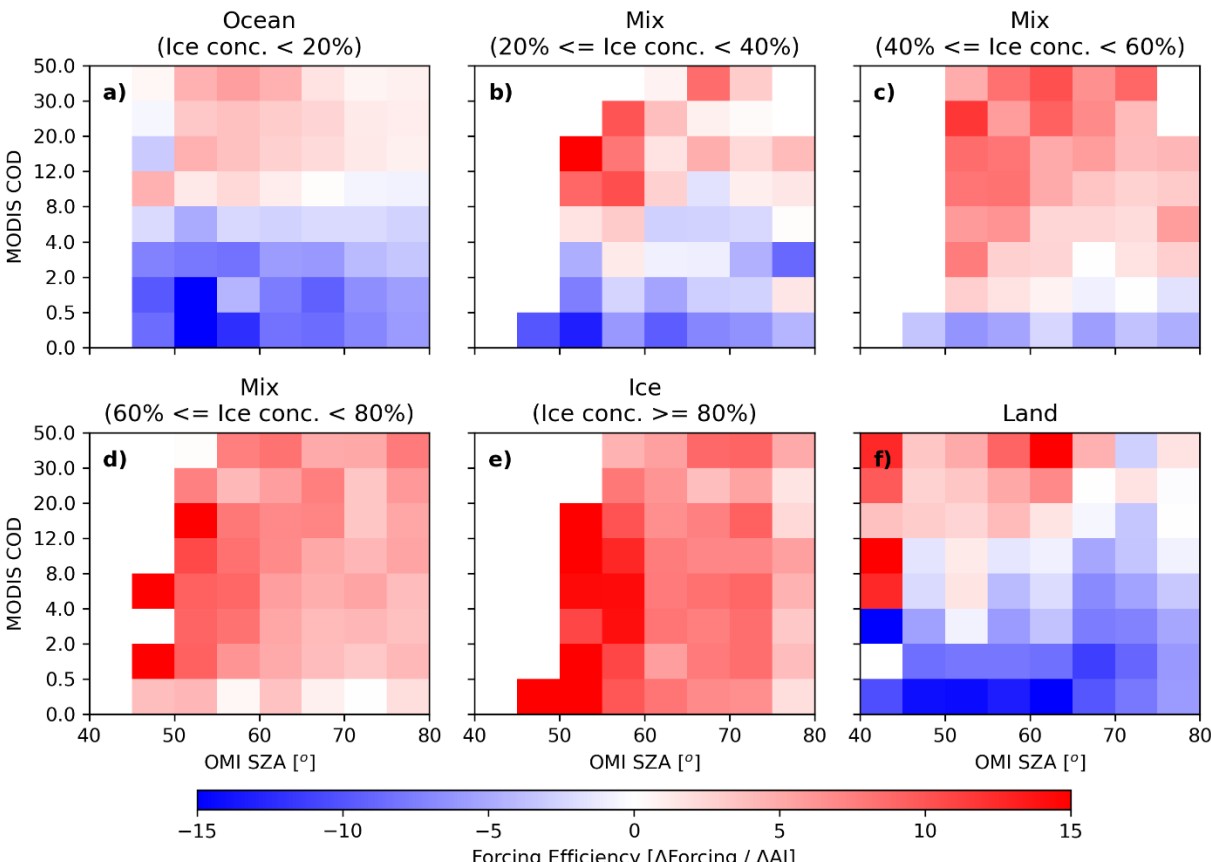

**Figure 9.** Slopes of the regression lines between the UVAI perturbations and NN-based ADRF estimates as functions of OMI solar zenith angle and MODIS COD, for (a) ocean surfaces ( (a) sea ice concentration (SIC) below 20%), mixed ice/ocean surfaces ( (b) SIC between 20% and 40%, (c) SIC between 40% and 60%, (d) SIC between 60% and 80%), ice surfaces ( (e) SIC greater than 80%), and (f) land surfaces .

## 4.2 Calculate daily estimates of ADRF from LUT & daily-averaged OMI UVAI data

With the forcing efficiency values derived from the co-located L2 data, we then estimated ADRF on a daily basis from 1 April to 30 September of 2005 through 2020. Daily averages of perturbed OMI UVAI on a 1x1 degree latitude x longitude grid were derived from the QC-ed L2 OMI UVAI data, while L3 MODIS daily 1x1 degree gridded cloud optical depth (product MYD08_D3) were obtained from NASA Langley online data archive and daily SSMIS SICs on the default 25 x 25 km$^2$ grid were converted to a 1x1 degree latitude longitude grid. For each day, if a 1x1 degree OMI grid box contained a daily averaged perturbed OMI UVAI value that was higher than a threshold (here, set to 0.7), then a forcing value was estimated for that grid box. Regions with OMI UVAI values less than the threshold value were assumed to be aerosol free, and the ADRF values were set to zero for those regions. We note that similar results were obtained when, rather than using this straightforward threshold approach for the daily OMI UVAI data, we compared the daily OMI UVAI value to a UV-absorbing aerosol-free OMI background climatology value and

calculated daily forcing if the daily UVAI was greater than the background by more than the threshold amount. The daily SSMIS SIC value, the calculated daily minimum solar zenith angle, and the L3 MODIS COD values for that grid box were used to select the correct forcing regression slope and intercept from the forcing regression LUT. Once the forcing efficiency slope and intercept values were identified, the estimated daily ADRF was calculated following:

$$ADRF[i,j] = \frac{\partial ADRF}{\partial AI}\Big|_{ICE[i,j],SZA[i,j],COD[i,j]} \times UVAI[i,j] + C_{ADRF}\Big|_{ICE[i,j],SZA[i,j],COD[i,j]} \tag{3}$$

where $i$ denotes the latitude index, $j$ denotes the longitude index, $UVAI[i,j]$ is the daily UVAI for the grid box, and $\frac{\partial ADRF}{\partial AI}\Big|_{ICE[i,j],SZA[i,j],COD[i,j]}$ and $C_{ADRF}\Big|_{ICE[i,j],SZA[i,j],COD[i,j]}$ are the forcing efficiency slope and intercept, respectively, associated with the SIC, solar zenith angle, and COD of the lat/lon grid box. Thus, although both SIC and cloud coverage change throughout the study period, their combined impact to ADRF is reflected in the analysis. An example of the daily estimated aerosol forcing for 5 July 2018 is shown in Fig. 10. In Fig. 10a, the daily averages of perturbed

OMI UVAI reveal a large plume of BB smoke over northeastern Russia and extending over the Arctic Ocean. The SSMIS SIC (Fig. 10b) and MODIS COD (Fig. 10c) values indicate that most of this plume was located over primarily ocean and ice surfaces, with a mixture of cloudy and cloud-free conditions in those regions. After following the methodology described above, the daily estimated ADRF was calculated, with the forcing value being shown in Fig. 10d. The positive (red) values indicate less upward-directed SW energy caused by the BB aerosol particles, which is

expected due to the icy and cloudy conditions in those regions. The same plume also exhibited negative (blue) forcing values across the land and mixed ice/ocean areas.

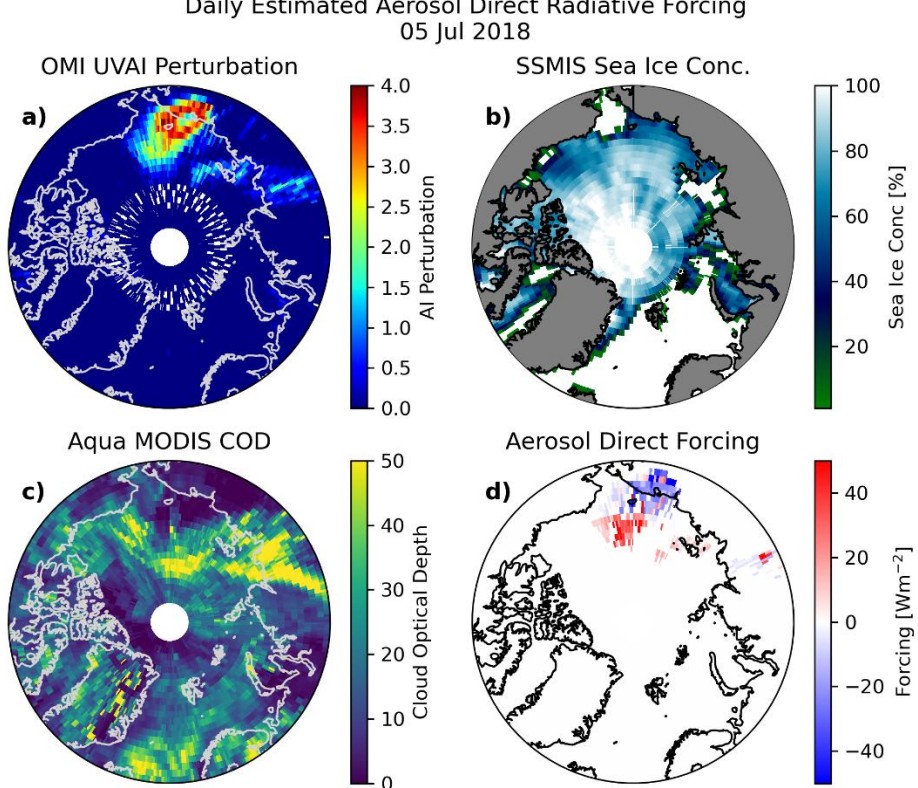

**Figure 10.** Estimated forcings from aerosol using forcing efficiency slopes and intercepts for 5 July 2018. a) Daily averaged perturbed OMI UVAI. b) SSMIS sea ice concentration (SIC). c) Daily L3 Aqua MODIS COD. d) Estimated aerosol direct radiative forcing (ADRF) for 5 July 2018 based on the OMI UVAI and the look-up table (LUT) of aerosol forcing regressions under different viewing geometry, surface, ad cloud conditions.

### 4.3 Error analysis of daily-estimated aerosol direct radiative forcing

Before applying the NN results and forcing regression LUT to long-term ADRF trend analyses, an analysis of the impacts of errors in the system on the daily observation-based estimates of ADRF must be conducted. Thus, we calculated error statistics for four error sources: errors in the neural network output, errors in the forcing regressions used in the LUT, impacts of daily SSMIS SIC errors on the daily forcing estimates, and impacts of daily MODIS COD errors on the daily forcing estimates.

First, we quantified the errors in the NN-generated aerosol-free SWF estimates against CERES observations. For each of the 50 L2 swaths reserved for testing (not involved in training the NN), aerosol-free SWF values were estimated and were compared to the CERES SWF observations from the same swaths. All these errors were combined, and after removing any pixels with UVAI > 1, the distribution of the combined errors from all 50 swaths was generated and is shown in Fig. 11a. The red curve represents a Gaussian curve fitted to the distribution, fitted using the Levenberg-Marquardt algorithm and least squares statistics. A normal distribution fits the errors well, though with a slight

underestimation of errors towards the edges of the distribution. Based on the fitted Gaussian curve, the mean and standard deviation of the NN output errors are 1.4 Wm$^{-2}$ and 18.3 Wm$^{-2}$, respectively (Fig. 11a).

    Another source of error in the daily estimates of ADRF is in the application of the forcing regressions in the LUT. To quantify errors in the LUT method at estimating ADRF, we first calculated ADRF for all of the aerosol swaths by subtracting the CERES observations from the NN aerosol-free SWF output; this is referred to as the "L2-style" forcing

estimate. Then, for the same L2 swaths, we calculated estimated ADRF at each aerosol-containing pixel using the LUT-based method, in which the MODIS COD, OMI SZA, and SSMIS SIC values were used to select the forcing regression values from the LUT, and the OMI UVAI perturbation from the L2 pixel was then applied to the forcing regression values to generate an estimated ADRF; this is referred to as the "L3-style" forcing estimate. The errors between the L2-style and L3-style forcing estimates for all aerosol-containing pixels in the L2 aerosol swaths were

combined, and the distribution of the combined errors is shown in Fig. 11b. As in Fig. 11a, a Gaussian curve was fitted to the data using the Levenberg-Marquardt algorithm and least squares, and the normal distribution provides a good estimate for the errors. The mean and standard deviation of the L2-style vs. L3-style errors are -1.3 Wm$^{-2}$ and 21.0 Wm$^{-2}$, showing that the L3-style ADRF values slightly overestimate the ADRF computed directly from the NN output and CERES observations.

Lastly, we investigated how errors in the daily L3 SSMIS SICs and MODIS COD values affect the daily estimated ADRF. According to the SSMIS daily SIC dataset user guide from the National Snow and Ice Data Center, and as described in Section 2.4, SSMIS SICs are generally within +/- 5% of the true SIC in the wintertime, and within +/- 15% in the summertime due to the presence of melt ponds on the ice surface. Thus, to determine how possible errors in the daily SSMIS SICs of this magnitude impact the estimated daily ADRF values, we first calculated daily estimated

ADRF using the methods described in Section 4.2 for 1 April to 30 September of 2005 – 2020. Then, we calculated the daily ADRF values again, but before using the daily SSMIS SIC to select the aerosol forcing regression values from the LUT, we perturbed the SIC by an error from a normal distribution with a mean of 0% and a standard deviation of 15%, though the ending SIC values were capped to a minimum of 0% and a maximum of 100% after adding the errors. The distribution of the errors between the original daily L3 ADRF estimates and the ice error-affected daily L3

ADRF estimates are shown in Fig. 11c; note that the y-axis is set to a logarithmic scale because the vast majority of the errors are equal to 0 (the "0" bin contains about 60,000 values while the next closest bins contain less than 1,000 values). We suspect that the overwhelming frequency of 0-value errors in the distribution is due to the chosen SSMIS surface bins and the wide coverage of land surfaces within the study area. With the SSMIS SIC bins used in the LUT being 20% wide, if the error applied to the daily SSMIS SIC value was too small to change the sea ice value to a

different sea ice bin, the forcing regression values selected from the LUT did not change, and therefore the calculated daily ADRF value remained unchanged from the original calculation. Also, these changes did not affect the calculations over land surfaces, and with the source region for the aerosol plumes in the Arctic primarily being boreal Russia and Canada, many of the identified smoky grid points were over land and were unaffected by the perturbations in the ice values. Unlike the previous two error distributions, the errors in Fig. 11c are not normally distributed. The

mean and standard deviation of the errors in daily L3 ADRF estimates due to SSMIS sea ice errors are 0 Wm$^{-2}$ and 3.2 Wm$^{-2}$, respectively.

Similar methods were applied to determine the impacts of errors in the daily MODIS COD values on the estimated daily ADRF values. After surveying the standard deviations of the L1B/L2 MODIS COD values that were averaged into each daily L3 COD value across the entire 1 April to 30 September of 2005 – 2020 dataset, we found that, though

the most commonly-occurring COD standard deviation is less than 1, the second most commonly-occurring daily MODIS COD standard deviation is about 5. Thus, similar to above, we recalculated the daily L3 ADRF values, but before using the MODIS COD value at the grid point to select the forcing regression values from the LUT, we perturbed the COD value by an error from a normal distribution with a mean of 0 and a standard deviation of 5. The distribution of the errors between the original ADRF values and the values calculated using the perturbed COD values

are shown in Fig. 11d; as in Fig. 11c, a logarithmic y-axis is used in Fig. 11d because the vast majority of the errors are equal to 0, with the "0" bin containing about 40,000 values while the next closest bins have about 8,000 values. We suspect that there are more non-zero errors in the COD error distribution than in the ice error distribution because of the small COD bin sizes for lower COD values. As with the ice errors, the COD-induced forcing errors are not normally distributed. The mean and standard deviation of the errors in daily L3 ADRF estimates due to MODIS COD

errors are 0.7 Wm$^{-2}$ and 14.6 Wm$^{-2}$, respectively.

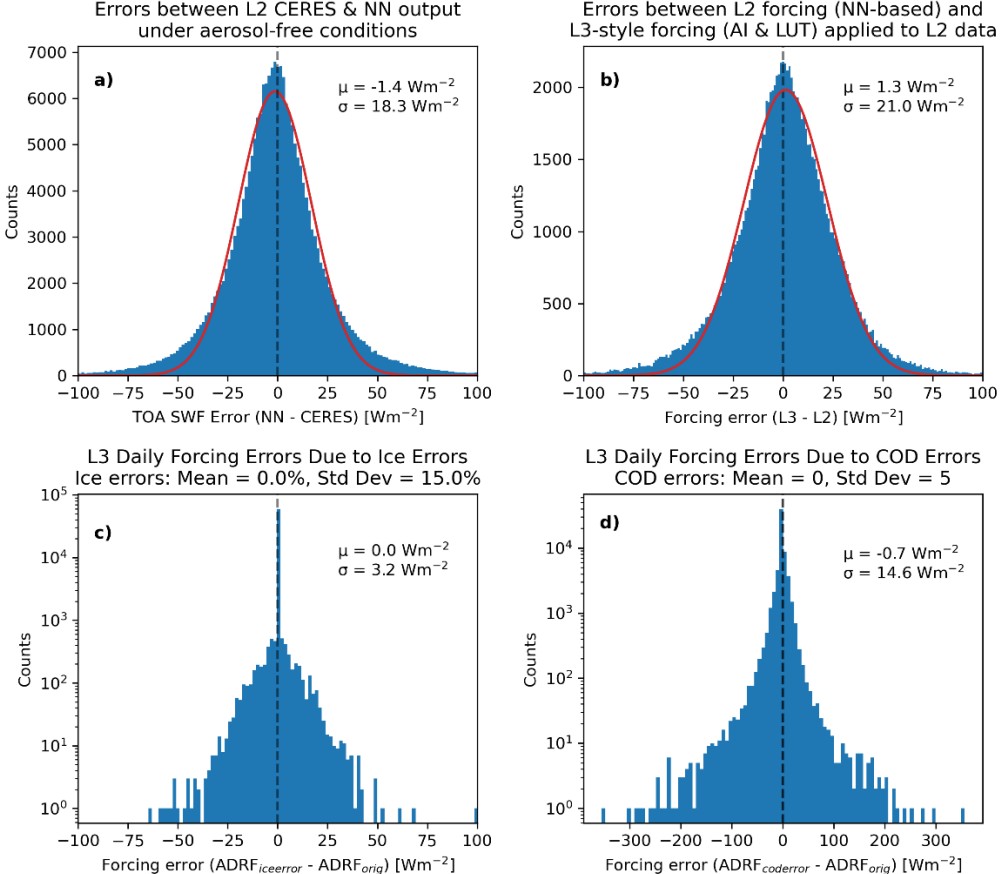

**Figure 11.** Error distributions of four error sources in the daily forcing estimates. a) Distribution of errors between the CERES SWF observations and the NN-estimated aerosol-free SWF, generated from the 50 reserved co-located L2 swaths and for pixels with UVAI < 1. b) Distribution of errors between the ADRF calculated directly from the NN output and CERES observations (L2-style) and the ADRF calculated on the L2 swaths using the OMI UVAI perturbations and the look-up table (LUT) of forcing regressions (L3-style). c) Distribution of errors in the daily L3 forcing estimates caused by the application of normally-distributed errors to the daily SSMIS sea ice values during the calculation process. d) Similar to panel c, but showing the impacts of the application of errors in the daily MODIS COD values. The mean and standard deviation of the distributions in Wm⁻² are represented by μ and σ, respectively, in each panel. Note that a logarithmic y-axis is applied to panels (c) and (d).

With the mean and standard deviation of the errors from each component calculated, the error statistics were combined to derive the total error statistics for the daily L3 observation-estimated ADRF values. The mean error was calculated as the sum of the individual error distribution means, which is -0.8 Wm⁻². This slight low bias can be corrected in the L3 daily ADRF calculations by adding 0.8 Wm⁻² to the calculated daily ADRF values. The final standard deviation was calculated as the square root of the sum of the squares of each individual standard deviation, which equates to 31.6 Wm⁻², assuming error variances are addable. We note that this does not account for co-varying errors (such as related errors in COD and SIC), so this standard deviation is likely an overestimate of the true error in the daily forcing values.

**4.4  Total observation-based Arctic aerosol forcing trends using Monte Carlo error estimations**

We conducted a Monte Carlo simulation method to estimate the impact of the errors in the observational estimates of daily L3 ADRF. In this approach, a total of 600 independent simulations were performed. We note that, although 600 simulations were selected, the mean trend values largely stabilized after 300 simulations, with a less than 5% difference found between mean ADRF trends larger than 0.25 $Wm^{-2}$ per study period for simulations with 300 and 600 runs. For each simulation, daily ADRF values were computed from the LUT as mentioned in Section 4.2 using

daily OMI UVAI, SIC and COD data at a spatial resolution 1x1° latitude/longitude over the study domain. For each day and each 1x1° latitude/longitude grid, an error in ADRF was added to the daily ADRF value. The added error term was randomly generated by following the normal distribution as derived from Section 4.3, with a mean of 0.8 $Wm^{-2}$ and a standard deviation of 31.6 $Wm^{-2}$. For each simulation, ADRF trends (2005-2020) can be estimated, first by averaging daily values into monthly averages, and then by estimating trends through the linear regression analyses

over the monthly averages. Since we added semi-random errors to the ADRF calculations, with errors added following the accumulated error distributions from Section 4.3, in theory, with sufficient simulations, the spread of aerosol forcing trends from those simulations shall capture error sources as mentioned in Section 4.3 (e.g. with an error standard deviation of 31.6 $Wm^{-2}$). Here we assumed that the errors in the daily L3 ADRF values are normally distributed. We note that the same number of data points were included in each simulation, and the only difference

among the 600 simulations is that for each observation for each simulation, the added error term, which was randomly generated based on the error distribution from Section 4.3, was different.

The mean trend from the 600 simulations was considered the ADRF trend from this study and the spread in trends from different simulations was related to the error boundaries of the calculated ADRF trend. This exercise was performed for April – September for the study period of 2005-2020. In addition to computing the trends of ADRF

over the study period, we computed monthly trends of SSMIS SIC, Aqua MODIS cloud fraction, and perturbed OMI UVAI data over the study period for qualitative comparison with the ADRF trends. The monthly SSMIS SIC values were first averaged into a 1 x 1° latitude/longitude grid, and then linear regression was applied to the time series of monthly averaged SIC at each grid point to find the trends. Linear regression was also applied to the L3 monthly Aqua MODIS cloud fraction data at each latitude/longitude grid point to find cloud trends. Monthly perturbed UVAI values

were calculated by averaging together all daily UVAI averages that were greater than 0, and linear regression was applied to the time series of monthly perturbed UVAI data at each grid point to find the trends.

To determine the significance of the ADRF trends, we analyzed the mean and the spread of the 600 estimated monthly trends at each lat/lon grid point. We used the standard deviation of the 600 trends to construct a 90% confidence interval around the mean of each trend, and if the bounds of the confidence interval were the same sign as the mean

trend (i.e. if the absolute value of the trend was greater than 1.645 times the standard deviation of the 600 trends), we denoted the trend as being significant. An example of applying this methodology for the 600 trends at a point over northern Russia is shown in Fig. 12. The histogram of all 600 trend estimates at that point reveal that the trends are centered just above -4 $Wm^{-2}$ per study period, though some of the estimates are as low as -7 $Wm^{-2}$ per study period

and some are as high as about 0 Wm$^{-2}$ per study period. The dashed black line represents the mean of the 600 trends,
which is equal to -3.7 Wm$^{-2}$ per study period, while the dotted black lines represent the 90% confidence interval. With
a standard deviation of 1.3 Wm$^{-2}$ per study period, the confidence interval of the trend of "$\mu \pm 1.645\sigma$" becomes (-5.9, -1.6), and since this interval does not contain the value of zero, we denote this trend as being significant at the
90% confidence level.

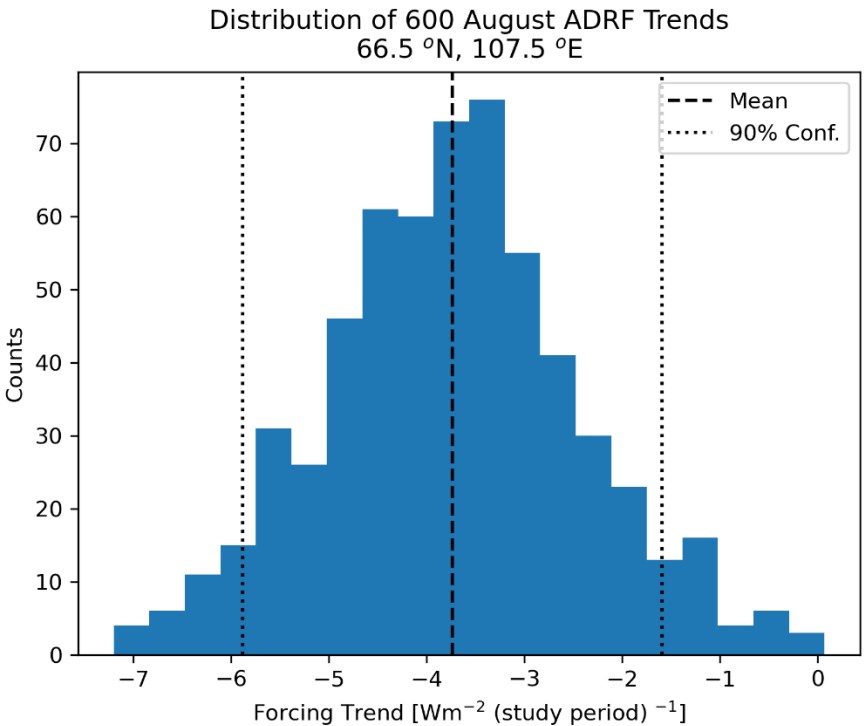

**Figure 12.** Example of the methods applied for determining the significance of the observation-based estimates of ADRF trend at each lat/lon grid point in the Arctic. The histogram shows the distribution of the 600 trend estimates at the selected point. The black dotted line represents the mean trend value, while the dotted black lines on either side of the mean represent the mean plus and minus the standard deviation.

The monthly trends of SSMIS SIC, Aqua MODIS cloud fraction, OMI UVAI, and the mean and standard deviation
of the 600 observation-based ADRF trends are shown in Fig. 13, with trends calculated separately per month.
Decreases in SIC (Fig. 13, first column) are strongest in the late summer months and September, times of the year
when Arctic sea ice extent is at its yearly minimum. The Aqua MODIS cloud fraction trends (Fig. 13, second column),
though variable by month and region, are largely positive across the Arctic, though decreases in cloud fraction were
found over Russia in June and August and over Europe in July.

Both the UVAI (Fig. 13, third column) and ADRF trends (Fig. 13, fourth column) are weak in the spring months, with
weak negative UVAI trends over much of the Arctic. Weak positive ADRF trends are found over parts of Russia and
Alaska, with regions of weak negative ADRF trend over the Arctic Ocean north of Alaska and Russia. Summertime
UVAI and ADRF trends, in contrast, are much stronger. Positive UVAI trends are found over Russia for June, July,
and August, as well as over Canada in August, though a region of negative UVAI trend over northwestern Russia can

be attributed to a large-scale BB aerosol event in that area early in the study period. Weaker UVAI trends extend from Russia over the Arctic Ocean. The ADRF trends for the summer months largely follow the patterns in the UVAI trends, with strong decreases in ADRF over Russia in June, July, and August, and over northern Canada in August. The negative ADRF trends are as large as -4 Wm$^{-2}$ per study period locally. These results indicate stronger ADRF closer to the BB aerosol source regions in north central Russia and Canada, with weaker ADRF for the BB smoke

plumes after being transported. Over the Arctic Ocean, in regions where BB aerosols are transported from mainland Russia and Canada, the magnitudes of the ADRF trends are smaller, and both negative and positive ADRF trends are found in different areas of the Arctic. Positive ADRF trends are found over the Arctic Ocean north of Russia and Alaska in July, as well as north of Canada in August, while negative ADRF trends are found closer to the northern coasts of Russia, Canada, and Alaska. The strongest positive ADRF trends over the Arctic Ocean, north of Russia in

July, are as high as +1 Wm$^{-2}$ per study period. We determined the confidence of the trends at each grid point using the methods described above, and trends in which the 90% confidence interval is nonzero are denoted with black hashing in Fig. 13. Most of the April and May ADRF trends do not have high confidence, but many of the strong negative ADRF trends over Russia and Canada in June, July, and August have high confidence. The positive trends over the Arctic Ocean north of Russia in July and north of Canada in August also have high confidence, but most of the other

weak trends over the Arctic do not. We suspect that the lower confidence in the weaker trends across much of the Arctic is likely a result of the high errors in daily observation-based estimates of ADRF.

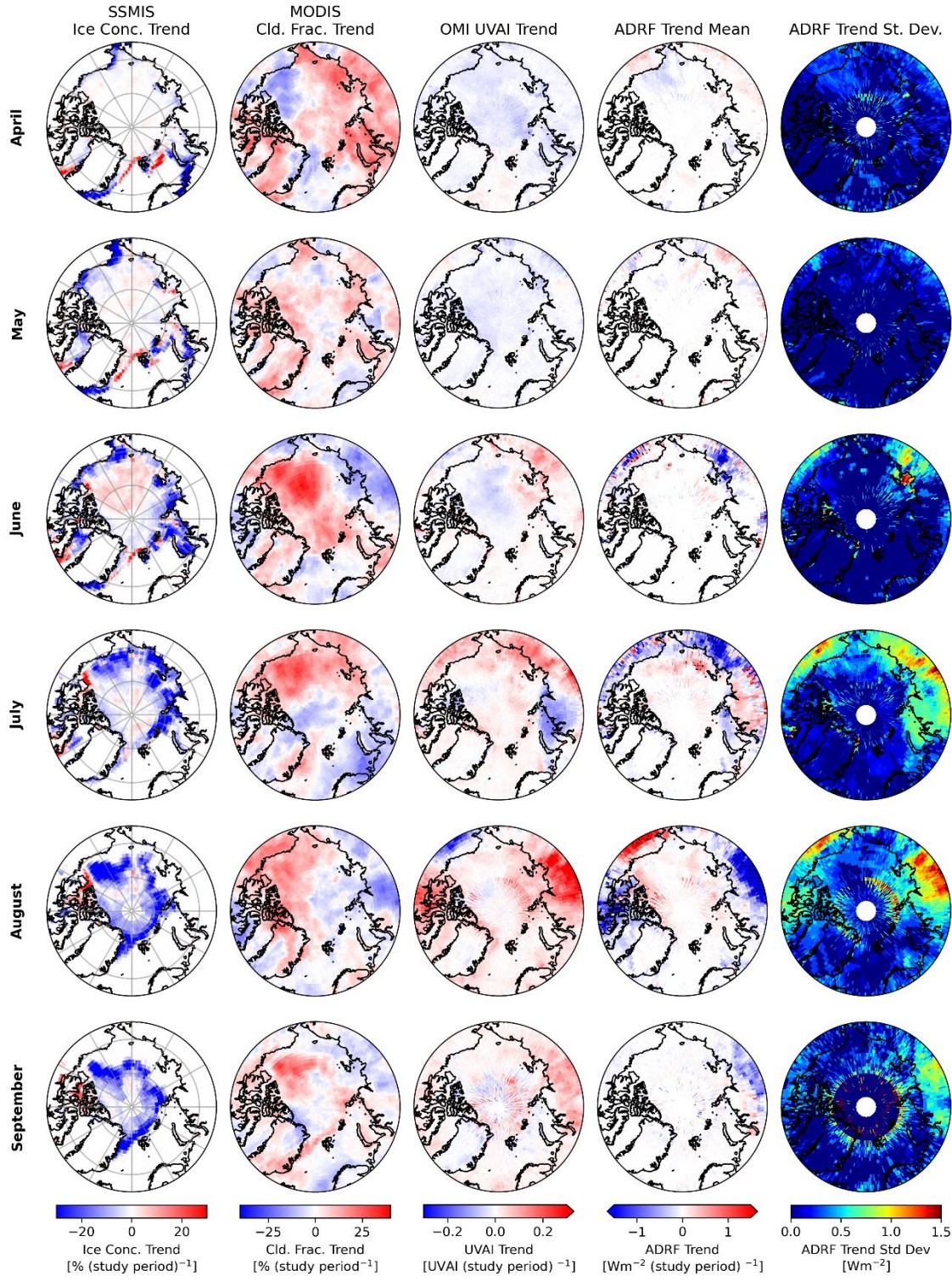

**Figure 13**. Monthly trends in SSMIS sea ice concentration (SIC), Aqua MODIS cloud fraction, OMI UVAI and observation-estimated ADRF trends over the 2005 – 2020 study period for April (1st row), May (2nd row), June (3rd row), July (4th row), August (5th row), and September (6th row). First column: Trends of monthly-averaged SSMIS SIC. Second column: Trends of monthly-averaged Aqua MODIS cloud fraction. Third column: Trends of monthly-averaged perturbed OMI UVAI, Fourth column: Mean

of 600 trends from monthly averages of observation-based ADRF estimates, plotted in units of Wm$^{-2}$ per study period. Fifth column: Standard deviation of the 600 monthly ADRF trends, plotted in units of Wm$^{-2}$ per study period.

In addition to the spatial trends, we analyzed the regional averages of observation-based ADRF estimates across the Arctic. We calculated regional averages of the 600 monthly forcing values in three regions: the entire Arctic (65° N – 90° N), low Arctic (65° N – 75° N), and high Arctic (75° N – 90° N). Then, we calculated the mean and standard deviation of the 600 trends across those regional averages of ADRF for each region. The mean trends are considered statistically significant if the mean of the p values associated with the 600 trends is below 0.05. The mean and standard deviation of the monthly mean and trends across the 600 trends are listed in Tab. 4, with bolded trends denoting those that are statistically significant. We note that similar results were obtained when using 300 trend estimates. Trends in region-averaged ADRF for the spring months are very small for all three regions. Larger trends are found in the summer months, with the low Arctic having the largest trends. Over the entire Arctic region, the strongest trends in regional observation-based ADRF estimates are in August, with a trend of -0.059 +/- 0.005 Wm$^{-2}$ per study period. Over the low Arctic, however, the trend is much larger at a statistically significant -0.185 +/- 0.009 Wm$^{-2}$ per study period for August. Trends in the high Arctic are the smallest, with the largest trends in the high Arctic found in August at +0.046 +/- 0.006 Wm$^{-2}$ per study period. While some of the trends in region-averaged ADRF are statistically significant, we admit that statistically significant trends may not actually be impactful. However, the magnitudes of Arctic annual mean aerosol radiative forcing estimates from model-based studies (Breider et al., 2017; Feng et al., 2013; Markowicz et al., 2021; Myhre et al., 2013; Schacht et al., 2019) range from 0.05 Wm$^{-2}$ to 0.64 Wm$^{-2}$ . Thus, some of the larger trends in monthly regional ADRF averages, such as those calculated for the low Arctic in the boreal summer, are comparable in magnitude to regional mean forcing estimates and therefore may be impactful.

**Table 4.** Mean and standard deviation of the trends over the 600 region-averaged monthly forcing estimates, separated by month, for 2005 – 2020, in units of Wm$^{-2}$ per study period. The uncertainty range denotes the standard deviation of the 600 trend estimates. Bolded trends are statistically significant, with the mean of the p values associated with the 600 trends being below 0.05.

|     | Arctic (65° N – 90° N) | Low Arctic (65° N – 75° N) | High Arctic (75° N – 90° N) |
| --- | --- | --- | --- |
| **Apr** | -0.002 +/- 0.002 | 0.002 +/- 0.003 | -0.005 +/- 0.003 |
| **May** | 0.001 +/- 0.001 | 0.004 +/- 0.002 | -0.001 +/- 0.002 |
| **Jun** | **-0.023 +/- 0.003** | **-0.053 +/- 0.005** | -0.003 +/- 0.003 |
| **Jul** | -0.031 +/- 0.004 | -0.078 +/- 0.008 | 0.008 +/- 0.004 |
| **Aug** | -0.059 +/- 0.005 | **-0.185 +/- 0.009** | 0.046 +/- 0.006 |
| **Sep** | -0.019 +/- 0.007 | -0.033 +/- 0.005 | -0.006 +/- 0.012 |

Changes in ADRF across the study period could result from changes in aerosol amount (UVAI) or from changes in the lower boundary condition (ice and cloud). First, to determine the impacts of sea ice change on the ADRF trends, we recalculated the daily ADRF estimates across the April – September of 2005 – 2020 study period while keeping the sea ice values unchanged from 2005. For example, in this method, the 1 August 2016 daily ADRF value was

calculated using the OMI UVAI and MODIS COD data from 1 August 2016, but with the SSMIS SIC data from 1 August 2005. We repeated this analysis while holding the ice values constant from other years and compared the trends in Arctic-averaged ADRF from the original calculations with those from the calculations with sea ice held constant. Overall, the trends calculated when holding the sea ice constant did not significantly vary from the initial trend results, with the mean percent difference over the 6 months being about 12%. We conducted a similar analysis

to determine the impacts of clouds on the trends in Arctic-averaged ADRF trends by recalculating the daily ADRF estimates across the study period while holding the cloud optical depth values unchanged from individual years. The trends calculated using the modified cloud optical depth values exhibited deviated more from the originals than the trends calculated with modified sea ice values, with a mean percent difference over the 6 months of about 65%. However, the signs of the monthly average forcing values and forcing trends remained largely the same, showing that

changes in UVAI were still the dominant factor causing the changes in absorbing aerosol direct radiative forcing.

## 5. Conclusions

In this study, through the use of satellite data from MODIS, CERES, SSMIS and OMI, we developed an observation based estimation of aerosol direct radiative forcing (ADRF) patterns and trends over the Arctic region for UV-absorbing aerosols for the period of 2005-2020. To derive ADRF, aerosol free sky TOA upwelling SW flux values

were derived through a neural network based method. Error distributions from various error sources were analyzed and an innovative Monte Carlo error estimation method was developed and implemented for quantifying uncertainties in estimated ADRF trends. This study found:

1. High $R^2$ values of above 0.9 were found between co-located CERES SWF data and the aerosol-free SWF values derived from a neural network-based method with the use of level 2 OMI, MODIS, SSMIS, and

CERES data as input parameters. The mean squared error (MSE) and mean absolute error (MAE) of 16.9 $Wm^{-2}$ and 2.86 $Wm^{-2}$, respectively, were found for the neural network after training based on aerosol-free SWF values, suggesting that the neural-network based method may be used for estimating aerosol-free SWF values for future aerosol forcing studies using CERES data.

2. With the combined use of OMI, CERES data, and with the use of aerosol-free SWF values as estimated from

the neural network-based method for over 130 aerosol-containing swaths over the Arctic, we quantified the instantaneous ADRF of absorbing aerosols (primarily BB) over the Arctic region as functions of solar zenith angle, surface type, and cloud conditions. For primarily cloud-free scenes (COD < 0.5) and with 20%-wide SSMIS SIC bins, a SIC of about 60% represents the turning point between SIC over which the scattering effects of BB aerosols ("cooling" effect) dominate to SIC over which the absorbing effects of BB aerosols

("warming" effect) dominate, though the ADRF over mixed ice/ocean surfaces is still rather mild due to lack of albedo contrast between the aerosol particles and the surface beneath. We note that a similar threshold of 60% - 65% is still found when using a variety of other SIC bin sizes and bin edges. Over primarily sea ice scenes and cloud-free conditions, instantaneous ADRF values can be as high as +80 $Wm^{-2}$ for heavy aerosol

loading (AI perturbation > 6), and over open water or over dark land, ADRF can be as low as -100 Wm$^{-2}$ for similar heavy aerosol loading scenarios.

3. To reduce computational burden, LUTs of ADRF as a function of observing conditions were constructed and were used to study long-term trends in observation-based ADRF at the OMI UVAI domain. The overall error in estimated daily ADRF, quantified as a Gaussian distribution, has a mean error of 0.8 Wm$^{-2}$ with a standard deviation of 31.6 Wm$^{-2}$. An innovative Monte Carlo method was introduced to estimate ADRF trends and uncertainties based on the daily ADRF error distribution, by introducing daily ADRF errors to the trend estimates through a stochastic-based method. As suggested from this study, strong negative ADRF trends as large as -4 Wm$^{-2}$ per study period were found over Russia and Alaska in the summer months, closer to the source region for the BB aerosols, with weaker trends over the Arctic Ocean. The trends over the Arctic Ocean in the boreal summer are mixed in sign, with both negative and positive ADRF trends found locally across the Arctic. The positive trends, which are generally closer to the North Pole than the negative trends, are as high as +1.0 Wm$^{-2}$ per study period in some regions.

4. When analyzing trends in regional averages of the monthly ADRF estimates around the Arctic, the strongest (and statistically significant) ADRF trends were found in the low Arctic (65 $^{\circ}$N – 75 $^{\circ}$N) at August at -0.185 +/- 0.009 Wm$^{-2}$ per study period. Trends in averaged ADRF over the high Arctic (75 $^{\circ}$N – 90 $^{\circ}$N) are much smaller than in the low Arctic, also peaking at August with a slightly positive and statistically insignificant trend of +0.046 +/- 0.006 Wm$^{-2}$ per study period.

This study suggests that while overall all changes in ARDF over the Arctic region are marginal and are only significant over certain period (e.g. August for the low Arctic), changes in regional ARDF can be significant and could contribute to regional warming and cooling and possible change in sea ice status, although ADRF over the Arctic region can be significantly affected by the underlying complex surface conditions. As the Arctic continues to warm, sea ice coverage continues to decrease, and intrusions of large amounts of BB smoke aerosol particles into the Arctic region become more frequent, these results suggest that absorbing aerosols may act to counter Arctic warming. This is still complicated, however, by Arctic sea ice and changes in Arctic cloud status, besides aerosol-cloud and aerosol-cryosphere interactions. Increases in Arctic cloud cover, especially in regions of sea ice loss (Abe et al., 2016), could mask the dark, ice-free ocean surfaces beneath the clouds and reduce the TOA cooling effect of lofted BB aerosol particles in the Arctic. However, the optical depth of the clouds over ocean surfaces have a significant impact on the TOA radiative forcing characteristics of a BB aerosol plume, so estimating how future changes in cloud status could affect future ADRF is very complicated. Additionally, BB smoke plumes reaching the high Arctic over sea ice regions may lead to local warming effects. Further, this study focused only on the direct radiative impacts of absorbing aerosols, leaving out the impacts of scattering aerosols. While scattering aerosols such as sea salt and sulfates have radiative cooling effects, reductions in sulfate emissions have led to decreases in sulfate aerosols. Thus, the cooling effects of sulfate aerosols is projected to weaken in the future (Ren et al., 2020; Schmale et al., 2022).

While we identified that the TOA radiative impacts of a lofted plume of absorbing aerosol particles switch from cooling (i.e. scene brightening) to warming (i.e. scene darkening) above a critical SIC threshold of 60% - 65%, this

raises several questions that are unanswered in this study. We do not know precisely why 60% – 65% represents the

critical threshold. Additionally, we do not know how other phenomena, such as multiple scattering between the aerosol

layer and the ice surface, impact the TOA forcing characteristics. Studies to investigate such questions would require

extensive radiative transfer model simulations using varying SIC, atmospheric temperature and moisture profiles,

cloud properties, and aerosol properties (with observations needed to quantify the aerosol properties over the multiple

surface types in the Arctic), which would go beyond the scope of this study. These are very interesting research

questions that warrant further study, but we leave them to future work.

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

**Author contributions**

J. Zhang and B. T. Sorenson designed the concept of the study. B. T. Sorenson implemented the study. J. S. Reid and P. Xian provided support and comments for the study. All authors contributed to writing and editing the manuscript.

**Competing interests**

The authors claim no competing interests.

**Financial support**

This project is supported by NASA Grant 80NSSC20K1260. Coauthor JSR was supported under a grant from the Office of Naval Research Code 322.

**Acknowledgements**

This work used advanced cyberinfrastructure resources provided by the University of North Dakota Computational Research Center.

**Appendix**

**Table A1.** The dates and visual description of the BB aerosol events from which L2 data were obtained and used in the study.

| Date | Event Description |
|---|---|
| 24-27 July 2006 | Large smoke plume from central Russia extending to the Arctic Ocean |

| 22 April 2008 | Smoke from Alaska extending north over the Arctic Ocean |
|---|---|
| 11-12 August 2014 | Smoke from NE Russia extending over Arctic ocean and sea ice |
| 27 June 2015 | Smoke over Alaska and the Beaufort Sea |
| 6 – 10 June 2015 | Yellow smoke over sea ice in Arctic ocean north of Alaska |
| 14-17 August 2017 | Smoke from pyrocumulonimbus event in British Columbia extended into the central Arctic |
| 3 – 5 July 2018 | Smoke from NE Russia crossing the Chukchi Sea and entering Alaska |
| 21 July 2018 | Large amounts of dark smoke over the Arctic between NE Siberia and Alaska |
| 14 August 2018 | Large smoke plume over Arctic Ocean (both ice and water) starting from central Siberia |
| 26 August 2018 | Large smoke plume across northern Canada and Greenland |
| 10 – 11 August 2019 | Smoke from NE Siberia lofted across the Arctic Ocean |

**Table A2.** The linear regression equations relating the UVAI and ADRF values binned by SSMIS surface type and MODIS COD from Fig. 8.

| **ADRF vs UVAI Regression Equations** Obtained from ADRF and UVAI Binned by SSMIS Sea Ice Concentrations and MODIS COD | | | | |
|---|---|---|---|---|
| | | **SSMIS Surface Type Ocean (0 – 20% ice)** | **SSMIS Surface Type Mix (20 – 40% ice)** | **SSMIS Surface Type Mix (40 – 60% ice)** |
| **MODIS COD** | **0 – 0.5** | ADRF = -11.4 * UVAI – 14.2 | ADRF = -8.3 * UVAI - 4.6 | ADRF = -4.8 * UVAI - 3.3 |
| | **0.5 – 2** | ADRF = -7.4 * UVAI – 7.4 | ADRF = -3.6 * UVAI + 0.2 | ADRF = 0.8 * UVAI - 1.3 |
| | **2 – 4** | ADRF = -6.3 * UVAI – 3.0 | ADRF = -1.9 * UVAI + 1.4 | ADRF = 2.3 * UVAI - 1.1 |
| | **4 – 8** | ADRF = -2.6 * UVAI – 1.8 | ADRF = -0.6 * UVAI + 1.3 | ADRF = 3.6 * UVAI - 2.6 |
| | **8 – 12** | ADRF = 0.9 * UVAI – 1.3 | ADRF = 3.8 * UVAI + 0.1 | ADRF = 5.8 * UVAI - 1.6 |
| | **12 – 20** | ADRF = 2.9 * UVAI – 1.9 | ADRF = 5.8 * UVAI - 1.9 | ADRF = 6.1 * UVAI - 3.0 |
| | **20 – 30** | ADRF = 2.7 * UVAI – 0.6 | ADRF = 3.0 * UVAI + 3.1 | ADRF = 7.4 * UVAI - 0.9 |
| | **30 – 50** | ADRF = 2.3 * UVAI + 2.3 | ADRF = 2.3 * UVAI + 3.1 | ADRF = 7.7 * UVAI + 1.2 |
| | | **SSMIS Surface Type Mix (60% – 80% ice)** | **SSMIS Surface Type Ice (80% – 100% ice)** | **SSMIS Surface Type Land** |
| **MODIS COD** | **0 – 0.5** | ADRF = 2.1 * UVAI - 5.2 | ADRF = 7.8 * UVAI + 1.6 | ADRF = -14.3 * UVAI - 5.4 |
| | **0.5 – 2** | ADRF = 5.9 * UVAI - 5.0 | ADRF = 8.2 * UVAI + 1.0 | ADRF = -8.4 * UVAI - 5.8 |
| | **2 – 4** | ADRF = 5.9 * UVAI - 4.0 | ADRF = 8.5 * UVAI - 2.6 | ADRF = -4.5 * UVAI - 6.8 |
| | **4 – 8** | ADRF = 6.2 * UVAI - 3.1 | ADRF = 8.5 * UVAI - 0.5 | ADRF = -2.2 * UVAI - 5.3 |
| | **8 – 12** | ADRF = 6.2 * UVAI - 1.3 | ADRF = 8.1 * UVAI + 1.5 | ADRF = -1.0 * UVAI - 3.5 |
| | **12 – 20** | ADRF = 7.1 * UVAI - 2.5 | ADRF = 9.1 * UVAI + 1.0 | ADRF = 2.3 * UVAI - 7.1 |
| | **20 – 30** | ADRF = 6.1 * UVAI + 2.3 | ADRF = 7.6 * UVAI + 2.7 | ADRF = 3.4 * UVAI - 6.4 |
| | **30 – 50** | ADRF = 6.2 * UVAI + 3.9 | ADRF = 7.3 * UVAI + 3.3 | ADRF = 5.3 * UVAI - 5.8 |