# Peer review of "An observational estimate of Arctic UV-absorbing aerosol direct radiative forcing on instantaneous and climatic scales"

_EGUsphere, 2025_

## Author Comment (AC1)

**Reviewer #1 Responses**

*The authors use a data-driven approach (based on long-term satellite observations + neural networks + Monte Carlo methods) to study the impact of Arctic sea ice cover on absorbing aerosol direct radiative forcing (ADRF) and reveal its long-term trend. The following issues should be addressed before publication:*

Response: we thank the reviewer for constructive comments.

1. *The study's methodology is not highly innovative, as NN + data has been used in previous studies. However, the scientific findings seem more novel, particularly regarding the impact of sea ice on ADRF and the long-term trend. The literature review should more clearly compare this work with existing studies to confirm the novelty of the sea ice-ADRF relationship, as many GCM studies have already estimated Arctic ADRF and explored aerosol-sea ice interactions.*

   *RESPONSE:* We thank the reviewer for the comments and suggestions. We have added the following text to the introduction:

   "Similarly, previous studies have investigated the interactions between aerosol particles and snow- and ice-covered surfaces, with many using global climate models to determine how the deposition of absorbing particles onto sea ice and snow impacts the aerosol-radiation interactions (Bond et al., 2013; Flanner et al., 2007; Gagné et al., 2015; Schacht et al., 2019; Shindell and Faluvegi, 2009). Some studies have even investigated how changes in sea ice coverage affect aerosol radiative forcing in the Arctic. Using a global climate model, Struthers et al. (2011) found that reductions in Arctic sea ice extent led to increased emissions of sea spray/salt aerosol particles, with the associated increase in total AOD leading to stronger aerosol radiative cooling effects and a negative feedback on the Arctic climate."

2. *The 50+ samples used to train the NN might not be sufficient to represent all atmospheric conditions in the Arctic fully. If the NN primarily learns radiation fluxes from low-aerosol regions while the studied region has aerosols at higher altitudes, SWFcln may have systematic bias, leading to ADRF estimation errors. Stronger independent validation is needed to ensure the reliability of NN predictions.*

*RESPONSE:* We thank the reviewer for the comments and suggestions. When counting the training data in terms of the number of swaths/granules, the training dataset seems limited. However, each swath/granule contains sufficient data points with different observing conditions and viewing geometries. The dataset used for training and validation during the training process for the NN consisted of colocated L2 pixels from over 1100 OMI swaths over the Arctic across the boreal sunlit months from 2005 – 2020. This equates to about 4.4 million pixels, and after reserving 10% of the pixels for testing purposes during the training process, nearly 4 million pixels were used to train the model while 400,000 pixels were used to test the model during training.

After training the model, we used 50 swaths of co-located L2 data (independent of the training dataset) to validate the trained model, equating to about 200,000 pixels. While the initial results of validating the NN against these 50 reserved L2 swaths showed little systematic bias, we further validated the model by randomly selecting an additional 25 aerosol-free OMI swaths from the 2005 – 2020 data record. We co-located CERES, MODIS, and SSMIS data to those OMI swaths, and then added those swaths to the post-training validation dataset to determine the model's performance in these other unknown conditions. The new validation dataset thus consisted of about 300,000 pixels. The results from validating the model against CERES SWF obs for the new dataset showed similar results to the initial validation, suggesting that the model is not significantly affected by systematic biases. As we show later in response to comment #4, binning the NN errors by SSMIS surface type and MODIS COD yields little overall systematic bias as a function of those variables. We added a new Section 3.2 titled "Validation of the NN against CERES" to discuss our investigation of potential systematic biases in the NN output.

3. *In the section on neural network design on pages 11-12 (lines 300-315), the authors mention: "All nodes in the hidden layer use the Leaky Rectified Linear Unit (LeakyReLU) activation (Maas et al., 2013), with this activation function having been identified to provide the best performance after testing with other activation functions." Why does LeakyReLU provide the best performance? It is suggested that comparative test results for different activation functions (such as ReLU, Sigmoid, etc.) be provided and that the specific advantages of LeakyReLU in handling TOA radiation flux estimation be explained.*

*RESPONSE:* We thank the reviewer for the comments and suggestions. To investigate how the LeakyReLU gives better performance than other activation functions, we retrained the model using many other activation functions and compared the mean absolute errors both during and at the end of training. The ending mean absolute errors (MAE) of the NN-predicted aerosol-free SWF against CERES SWF observations after training for 100 epochs are listed in the table below. The LeakyReLU and ReLU activation functions gave the best performance, with the ending MAE for the simulations with LeakyReLU and ReLU activation being 2.86 Wm$^{-2}$ and 2.92 Wm$^{-2}$, respectively. Other models that gave good performance, but slightly worse performance than LeakyReLU, include the softplus and softsign activation functions, though the simulation with the softplus activation function exhibited some instability between epochs 60 and 80. While the simulations with ELU and SELU activation functions ended with MAE of around 3.2 Wm$^{-2}$, the training was highly unstable, with the errors spiking randomly between 3.0 Wm$^{-2}$ and 3.5 Wm$^{-2}$ with each epoch. The linear activation function provided one of the worst performances with an ending MAE of 5.47 Wm$^{-2}$, while training with GELU and sigmoid were stopped early because the MAE after the first about 10 epochs remained at around 12 Wm$^{-2}$ and did not converge. Since the LeakyReLU activation function gave the best performance out of the other activation functions, we used this activation function in the NN hidden layers model during training. With the LeakyReLU activation function known to avoid the "dead neuron" problem associated with the ReLU activation function (Dubey et al., 2022; Maas et al., 2013), we suspect that this could be behind the slightly better performance of the LeakyReLU activation function than the ReLU activation function.

The following paragraph was added to the end of Section 3.1 to discuss the comparative test results of training the NN with different activation functions:

"Several experiments were conducted to determine the best activation function (AF) to use in the NN hidden layers. The NN was trained multiple times using different AFs in the hidden layer nodes, and the ending mean absolute errors (MAE) of the NN-predicted aerosol-free SWF against CERES SWF observations after training with each AF for 100 epochs are listed in Table 1. The Leaky Rectified Linear Unit (LeakyReLU, Maas et al., 2013) AF gave the best performance with an ending MAE of 2.86 Wm$^{-2}$, while the Rectified Linear Unit (ReLU, Nair and Hinton, 2010) AF gave the second-best performance with an ending MAE of 2.92 Wm$^{-2}$. With the LeakyReLU activation function known to avoid the "dead neuron" problem associated with the ReLU activation function (Dubey et al., 2022; Maas et al., 2013), we suspect that this could be behind the slightly better performance of the LeakyReLU AF relative to the ReLU AF. Other models that gave good performance, but slightly worse performance than LeakyReLU, include the softplus (Glorot et al., 2011) and softsign (Glorot and Bengio, 2010) AFs, though training with the softplus AF exhibited some instability between

epochs 60 and 80. While the experiments with Exponential Linear Unit (ELU, Clevert et al., 2016) and Scaled Exponential Linear Unit (SELU, Klambauer et al., 2017) AFs ended with MAE of around 3.2 Wm$^{-2}$, the training was highly unstable, with the errors spiking randomly between 3.0 Wm$^{-2}$ and 3.5 Wm$^{-2}$ with each epoch. The linear AF provided one of the worst performances with an ending MAE of 5.47 Wm$^{-2}$, while the training experiments with Gaussian Error Linear Unit (GELU, Hendrycks and Gimpel, 2016) and sigmoid AFs were stopped early because the MAE after the first about 10 epochs remained at around 12 Wm$^{-2}$ and did not converge. Since the LeakyReLU activation function gave the best performance out of the other activation functions tested in this experiment, we used this activation function in all NN hidden layer nodes during training. Training was conducted on a GPU node for 100 epochs with a batch size of 128, an Adam optimizer (Kingma and Ba, 2017), and with back-propagational loss being derived by minimizing the mean squared error. After training for 100 epochs, the mean squared error (MSE) and mean absolute error (MAE) of the model-estimated SWF values against the training observations were 16.9 Wm$^{-2}$ and 2.86 Wm$^{-2}$, respectively."

**Table 1. Mean absolute errors (MAE) of the neural network output after training for 100 epochs with several different activation functions. Training with the sigmoid and GELU activation functions was terminated after about 10 epochs due to the extremely high MAE and the lack of convergence during the training process.**

| Activation Function | Reference | Mean absolute error after training for 100 epochs (Wm$^{-2}$) |
|---|---|---|
| LeakyReLU | (Maas et al., 2013) | 2.86 |
| ReLU | (Nair and Hinton, 2010) | 2.92 |
| Softplus | (Glorot et al., 2011) | 2.94 |
| Softsign | (Glorot and Bengio, 2010) | 3.06 |
| ELU | (Clevert et al., 2016) | 3.21 |
| SELU | (Klambauer et al., 2017) | 3.32 |
| Tanh | | 4.87 |
| Linear | | 5.47 |
| Sigmoid | | ~12* |
| GELU | (Hendrycks and Gimpel, 2016) | ~12* |

4. *Monte Carlo methods quantify uncertainty but cannot verify potential systematic biases. How can the authors confirm that NN-predicted SWFcln has no systematic bias?*

*RESPONSE*: We thank the reviewer for the comment and question. To test for systematic biases in the neural network-based estimates of ADRF, which would lead to systematic biases in the daily L3 ADRF estimates, we binned the validation dataset first by the different surface types, and then by the different COD ranges to determine if systematic biases were associated with either variable. The NN error distributions binned by the SSMIS surface type and by the MODIS COD are shown in the figure below. We found that the mean SWF errors for all of these swaths are largely small, with magnitudes primarily less than 3 Wm$^{-2}$. The peaks of nearly all the error distributions for the different surface types and CODs are around 0, suggesting little systematic bias in the system overall associated with the different surface types and CODs. The mean error for the land distribution is slightly larger at -5.5 Wm$^{-2}$, suggesting a slight negative bias over land. We suspect that this is related to the lack of information about the land-based surface type in the system. For example, if the NN is primarily trained on dark land surfaces, but it is applied to brighter-than-normal land surfaces (e.g. snow- and ice-covered land), the NN will predict lower upwelling SWF than is actually seen by CERES. When excluding data from April and May from this analysis, the mean error for the over-land data is much smaller, supporting our hypothesis that the slight negative shift in the land-based error distribution is related to the land surface brightness that is unaccounted for in this system. Given that the majority of the smoke events analyzed in the study occurred in the summer months (June – August), we do not expect this potential low bias of the NN over bright land surfaces to significantly impact the results of our study.

We have added the following text and figure to the paper:

**3.2 "Validation of the NN against CERES**

Once trained, the NN was first applied to the 50 reserved aerosol-free validation swaths (independent from the 131 aerosol swaths) to validate the NN output against CERES observations. The 50 validation swaths contained about 200,000 pixels to use for validation; we note that similar validation results were obtained when increasing the size of the validation dataset to about 300,000 pixels by adding 25 additional aerosol-free OMI swaths (and co-located MODIS, SSMIS, and CERES data) randomly chosen from the 2005 – 2020 boreal summer study period. Errors were calculated between the NN-estimated aerosol-free SWF and the associated CERES TOA SWF observations, and the distribution of the errors from the 50 validation swaths is shown in **Figure 1**a. The error distribution peaks at about 0 Wm$^{-2}$, suggesting little overall bias in the NN-estimated aerosol-free SWF values. To further test for systematic biases in the NN-estimated aerosol-free SWF, we binned the validation dataset first by the SSMIS SIC and surface type, and then by MODIS COD. The NN error distributions binned by the SSMIS surface type and the MODIS COD are shown in **Figure 1**b and **Figure 1**c, respectively. We found that the mean SWF errors for the error distributions binned by SSMIS SIC and MODIS COD are largely small, with magnitudes primarily less than 3 Wm$^{-2}$. The peaks of nearly all the error distributions for the different surface types and CODs are around 0 Wm$^{-2}$, suggesting little systematic bias

in the system associated with the different surface types and CODs. The mean error for the land distribution (**Figure 1**b, brown) is slightly larger at -5.5 Wm[-2], suggesting a slight negative bias over land. We suspect that this is related to the lack of information about the land-based surface type in the system. If the NN is primarily trained on dark land surfaces, but it is applied to brighter-than-normal land surfaces (e.g. snow- and ice-covered land), the NN will predict lower upwelling SWF than is observed by CERES. When excluding data from April and May from this analysis, the mean error for the over-land data is much smaller, supporting our hypothesis that the slight negative shift in the land-based error distribution is related to the land surface brightness that is unaccounted for in this system. Given that the majority of the smoke events analyzed in the study occurred in the summer months (June – August), we do not expect this potential low bias of the NN over bright land surfaces to significantly impact the results of our study."

[Figure]

**Figure 1.** a) Distribution of errors in the neural network (NN)-estimated aerosol-free shortwave flux (SWF) relative to CERES TOA upwelling SWF observations for the 50 validation swaths reserved from the NN training dataset. b) As in (a), but with the errors binned by the SSMIS sea ice concentration (SIC) and surface type. c) As in (a), but with the errors binned by MODIS cloud optical depth (COD).

5.  *Can Figure 2 include quantitative data on misclassification, such as the percentage of smoke misidentified as clouds?*

*RESPONSE:* We thank the reviewer for the question. After colocating the MODIS L1B cloud mask data to the OMI data for the swath shown in Figure 2, we calculated the number of smoky OMI pixels (defined as OMI UVAI > 1.0) for which the colocated MODIS cloud type was incorrect (i.e. pixels that should be "clear" being classified otherwise). We found that the co-located MODIS L1B cloud type classification was incorrect for about 25% of the smoky OMI pixels in this case. We note that this

misclassification only happens for very optically dense smoke plumes, which are low probability events over polar regions. So, the number calculated from Figure 2 might not be representative. For this reason, we didn't add this number to Figure 2.

6. *Why does ADRF shift from negative (cooling) to positive (warming) at a critical sea ice concentration of approximately 60%? Why is 60% the turning point? How does aerosol-surface multiple scattering influence ADRF?*

*RESPONSE:* We thank the reviewer for the comments and questions. To further investigate the 60% sea ice concentration (SIC) threshold beyond which the ADRF shifts from negative (cooling) to positive (warming), we replotted the binned L2 UVAI vs ADRF results from Figure 7, but with several other SIC bin sizes and bin centers. For example, we rebinned the ADRF data using SIC bin widths of 5%, 10%, 15%, and 20%, and with bin edges of both 60% and 65% (in other words, SIC bins around the 60% critical threshold include 40% - 60%, 45% - 65%, 45% - 60%, 50% - 65%, 50 – 60%, 55 – 65%, etc.). The critical SIC threshold was found to be 60% in most of the recalculations, with the others having the threshold at the 65% bin. Thus, we are - confident that the critical SIC threshold is between 60% – 65%. We have modified the text in Section 4.1 to reflect this added confidence: "...positive between the 40% – 60% and 60% – 80% bins, or roughly a SIC of 60% *(we note that a similar threshold of 60% - 65% is also found when binning the ADRF data using a variety of other SIC bin sizes and bin edges)*." We have also added the following text to point 2 in the conclusions: "... though the ADRF over mixed ice/ocean surfaces is still rather mild due to lack of albedo contrast between the aerosol particles and the surface beneath. *We note that a similar threshold of 60% - 65% is still found when using a variety of other SIC bin sizes and bin edges*. Over primarily sea ice scenes..."

We are also curious as to why an SIC of 60 – 65% represents the turning point between TOA warming and cooling effects of lofted absorbing aerosol particles, and about the impacts of aerosol-surface multiple scattering on the ADRF results. In theory, this is related to the relative reflectance/scattering properties of the surface and aerosol layer (e.g. a bright aerosol layer over a darker surface versus a dark aerosol layer over a brighter surface). Such a study to investigate these phenomena would require extensive radiative transfer model simulations using varying sea ice concentrations, atmospheric temperature and moisture profiles, cloud properties, and aerosol properties (with observations needed to quantify the aerosol properties over the multiple surface types in the Arctic), which would go beyond the scope of

this study and make this study much longer than it already is. While this is a very interesting research question that warrants further study, we leave this to future work and simply report the identified critical threshold here. We have added the following paragraph to the end of the conclusions section:

"While we identified that the TOA radiative impacts of a lofted plume of absorbing aerosol particles change from cooling (i.e. scene brightening) to warming (i.e. scene darkening) above a critical SIC threshold of 60% - 65%, this raises several questions that are unanswered in this study. We do not know precisely why 60% – 65% represents the critical threshold. Additionally, we do not know how other phenomena, such as multiple scattering between the aerosol layer and the ice surface, impact the TOA forcing characteristics. Studies to investigate such questions would require extensive radiative transfer model simulations using varying sea ice concentrations, atmospheric temperature and moisture profiles, cloud properties, and aerosol properties (with observations needed to quantify the aerosol properties over the multiple surface types in the Arctic), which would go beyond the scope of this study. These are very interesting research questions that warrant further study, but we leave them to future work."

**References**

Bond, T. C., Doherty, S. J., Fahey, D. W., Forster, P. M., Berntsen, T., DeAngelo, B. J., Flanner, M. G., Ghan, S., Kärcher, B., Koch, D., Kinne, S., Kondo, Y., Quinn, P. K., Sarofim, M. C., Schultz, M. G., Schulz, M., Venkataraman, C., Zhang, H., Zhang, S., Bellouin, N., Guttikunda, S. K., Hopke, P. K., Jacobson, M. Z., Kaiser, J. W., Klimont, Z., Lohmann, U., Schwarz, J. P., Shindell, D., Storelvmo, T., Warren, S. G., and Zender, C. S.: Bounding the role of black carbon in the climate system: A scientific assessment, J. Geophys. Res. Atmospheres, 118, 5380–5552, https://doi.org/10.1002/jgrd.50171, 2013.

Clevert, D.-A., Unterthiner, T., and Hochreiter, S.: Fast and Accurate Deep Network Learning by Exponential Linear Units (ELUs), International Conference on Learning Representations, arXiv:1511.07289 [cs], https://doi.org/10.48550/arXiv.1511.07289, 2016.

Dubey, S. R., Singh, S. K., and Chaudhuri, B. B.: Activation functions in deep learning: A comprehensive survey and benchmark, Neurocomputing, 503, 92–108, https://doi.org/10.1016/j.neucom.2022.06.111, 2022.

Flanner, M. G., Zender, C. S., Randerson, J. T., and Rasch, P. J.: Present-day climate forcing and response from black carbon in snow, J. Geophys. Res. Atmospheres, 112, https://doi.org/10.1029/2006JD008003, 2007.

Gagné, M.-È., Gillett, N. P., and Fyfe, J. C.: Impact of aerosol emission controls on future Arctic sea ice cover, Geophys. Res. Lett., 42, 8481–8488, https://doi.org/10.1002/2015GL065504, 2015.

Glorot, X. and Bengio, Y.: Understanding the difficulty of training deep feedforward neural networks, in: Proceedings of the Thirteenth International Conference on Artificial Intelligence and Statistics, Proceedings of the Thirteenth International Conference on Artificial Intelligence and Statistics, 249–256, 2010.

Glorot, X., Bordes, A., and Bengio, Y.: Deep Sparse Rectifier Neural Networks, in: Proceedings of the Fourteenth International Conference on Artificial Intelligence and Statistics, Proceedings of the Fourteenth International Conference on Artificial Intelligence and Statistics, 315–323, 2011.

Hendrycks, D. and Gimpel, K.: Gaussian Error Linear Units (GELUs), https://doi.org/10.48550/arXiv.1606.08415, 2016.

Kingma, D. P. and Ba, J.: Adam: A Method for Stochastic Optimization, https://doi.org/10.48550/arXiv.1412.6980, 29 January 2017.

Klambauer, G., Unterthiner, T., Mayr, A., and Hochreiter, S.: Self-Normalizing Neural Networks, Neural Inf. Process. Syst. NIPS, 971–980, https://doi.org/10.48550/arXiv.1706.02515, 2017.

Maas, A. L., Hannun, A. Y., and Ng, A. Y.: Rectifier Nonlinearities Improve Neural Network Acoustic Models, in: Proceedings of the 30th International Conference on Machine Learning, International Conference on Machine Learning, Atlanta, GA, 3, 2013.

Nair, V. and Hinton, G. E.: Rectified linear units improve restricted boltzmann machines, in: Proceedings of the 27th International Conference on Machine Learning, International Conference on Machine Learning, Haifa, Israel, 807–814, https://doi.org/10.5555/3104322.3104425, 2010.

Schacht, J., Heinold, B., Quaas, J., Backman, J., Cherian, R., Ehrlich, A., Herber, A., Huang, W. T. K., Kondo, Y., Massling, A., Sinha, P. R., Weinzierl, B., Zanatta, M., and Tegen, I.: The importance of the representation of air pollution emissions for the modeled distribution and radiative effects of black carbon in the Arctic, Atmospheric Chem. Phys., 19, 11159–11183, https://doi.org/10.5194/acp-19-11159-2019, 2019.

Shindell, D. and Faluvegi, G.: Climate response to regional radiative forcing during the twentieth century, Nat. Geosci., 2, 294–300, https://doi.org/10.1038/ngeo473, 2009.

Struthers, H., Ekman, A. M. L., Glantz, P., Iversen, T., Kirkevåg, A., Mårtensson, E. M., Seland, Ø., and Nilsson, E. D.: The effect of sea ice loss on sea salt aerosol concentrations and the radiative balance in the Arctic, Atmospheric Chem. Phys., 11, 3459–3477, https://doi.org/10.5194/acp-11-3459-2011, 2011.

---

## Author Comment (AC2)

**Reviewer #2 Responses**

*This study presents an interesting data-driven approach, combining satellite observations, neural networks, and Monte Carlo uncertainty estimation, to derive Arctic absorbing aerosol direct radiative forcing (ADRF) trends over a 15-year period. The topic is highly relevant and timely, given the sensitivity of the Arctic climate. The manuscript is generally clear, but several areas require improvement to strengthen its scientific rigor and clarity.*

Response:  We thank the reviewer for constructive comments

1. *Data quality is critical for this analysis. Although the authors utilize several satellite products, many of these have primarily been validated over low- to mid-latitudes. Thorough validation over the Arctic region is necessary. More importantly, uncertainties associated with cloud, aerosol, and surface classification must be quantified. How reliable are the cloud-free and aerosol-free conditions as defined? Similar validation is needed for other aerosol and cloud products.*

   *RESPONSE:* We thank the reviewer for the comments. Regarding the quality of the surface classification dataset, we have added the following paragraph to Section 2.4 describing the accuracy of the SSMIS sea ice concentration dataset:

   "The SSMIS SIC dataset used in this study is one of two key SIC datasets provided by the NSIDC and has been used extensively in the scientific community to study Arctic sea ice trends. The algorithm used in the dataset, developed by NASA (Cavalieri et al., 1984), has been included in several SIC validation studies (Cavalieri et al., 1992; Ivanova et al., 2015; Kern et al., 2019, 2020; Meier, 2005; Steffen and Schweiger, 1991). Overall, and as reported in the NSIDC dataset user guide (https://nsidc.org/sites/default/files/documents/user-guide/nsidc-0051-v002-userguide.pdf), errors in the SIC dataset are less than 5% in the wintertime but can be as large as 15% in the summertime (Cavalieri et al., 1992). Some recent studies have reported that the SIC dataset may underestimate SIC by up to 10% (Kern et al., 2019, 2020), with the underestimation being partly caused by surface melt ponds in the summer months (Steffen and Schweiger, 1991). Additionally, microwave-based sea ice concentrations have been found to be sensitive to areas of thin ice (Ivanova et al., 2015). Nevertheless, despite some limitations, the algorithm and associated SIC dataset are widely used to represent Arctic SIC."

As for the MODIS cloud dataset used in the study, we note that additional checks are already included in MODIS cloud masking over polar regions, including the use of the observed radiance difference between the 6.7 and 11 μm channels. Still, there are known issues in cloud masking over the polar regions, especially associated with misclassification of snow and ice surfaces or smoke plumes as clouds (e.g. Fig. 2). It is for this reason that additional checks, such as the use of OMI AI and MODIS observations at 2.1 μm were included to mediate the issues (e.g. as shown in Fig. 2).

Regarding the quality of the aerosol information, we describe the methods by which we mitigate uncertainty in the OMI UVAI dataset in Section 2.1. Sorenson et al. (2023) developed a "perturbing method" to remove systematic biases and uncertainties in Arctic OMI UVAI data for use in quantitative Arctic aerosol studies. This method removes substantial viewing geometry and surface condition-related biases and uncertainties in the Arctic OMI UVAI data, and in addition to removing row anomaly-related uncertainty in the data following Sorenson et al. (2023), the OMI UVAI dataset is prepped for analysis in this study.

Regarding validation of the CERES data over the Arctic region, we have added the following paragraph to the end of Section 2.2 to describe the validation of CERES data in the Arctic region:

"CERES data have been used extensively for investigating changes in Arctic radiative energy budgets for both TOA (Duncan et al., 2020; Kay and L'Ecuyer, 2013; Riihelä et al., 2013) as well as the surface (Boeke and Taylor, 2016; Hegyi and Taylor, 2017). Previous studies have also worked to validate Arctic CERES surface radiative fluxes (Di Biagio et al., 2021; Riihelä et al., 2017) and TOA fluxes (Taylor et al., 2022), with the latter seeking to validate CERES TOA radiative fluxes against aircraft-based upwelling radiative flux observations. While Taylor et al. (2022) noted some error in the Arctic CERES Level-2 SSF TOA upwelling SWF resulting largely from errors in the imager-based sea ice concentrations used in the scene classification, the CERES observations compared well overall with the aircraft observations (differences between the CERES and aircraft observations were within $2\sigma$ uncertainty). The authors concluded that CERES TOA radiative flux data are suitable for polar climate studies (Taylor et al., 2022)."

2. *The authors should clarify whether all retrieved data were used or if any quality control measures (e.g., quality flags) were applied.*

*RESPONSE:* We thank the reviewer for the comment and suggestion. The quality control methods applied to the data are summarized in Table 1 in the paper. Following the Arctic OMI quality control methods described by Sorenson et al. (2023), we use the OMI row anomaly quality control flag to remove L2 OMI pixels from rows with known row anomaly issues. Then, we follow Sorenson et al. (2023) to remove additional row anomaly-affected rows that exhibit contamination over the Arctic. These quality control methods have cascading impacts on the rest of the datasets used in this study, as the L2 OMI UVAI data served as the basis for the co-location of the other satellite products. The surface type flag in the SSMIS sea ice concentration dataset was applied to remove pixels classified as "coastline" or as being too near the North Pole (i.e. in the "pole hole").

We have modified paragraph 2 of section 2.1 with the following text (new text is in italics here): "...with about 50% of the OMI rows currently being contaminated, so we *apply the row anomaly quality control flag in the OMI dataset to* exclude all flagged, row anomaly-affected rows from our analysis."

We also added the following text to paragraph 2 of Section 3 to describe the application of the quality control flags during the colocation process:

"As described in Section 2.1, we used the L2 OMI quality control flags and the methods described by Sorenson et al. (2023) to exclude pixels with flagged or unflagged OMI row anomaly contamination. from the colocated dataset. "

"We excluded pixels from the colocated dataset with the SSMIS surface type flag denoting coastline pixels or pixels too close to the North Pole (i.e. in the "pole hole")."

3. *The dataset used for training the neural network appears limited, which could introduce biases, particularly given the uncertain data quality. This needs careful discussion.*

*RESPONSE:* We thank the reviewer for the comments and suggestions. When counting the training data in terms of the number of swaths/granules, the training dataset seems limited. However, each swath/granule contains sufficient data points with different observing conditions and viewing geometries. The training dataset consisted of co-located L2 pixels from over 1100 OMI swaths over the Arctic across the boreal sunlit months of 2005 - 2020, equating to about **4 million pixels** for training and **400,000 pixels** for testing the neural network during the training process. The validation dataset used in the study consisted of about 50 independent swaths with a total size of about **200,000 pixels**.

To test for systematic biases in the neural network-based estimates of ADRF, we binned the validation dataset first by the different surface types, and then by the different COD ranges to determine if systematic biases were associated with either variable. The NN error distributions binned by the SSMIS surface type and by the MODIS COD are shown in the figure below. We found that the mean SWF errors for all of these swaths are largely small, with magnitudes primarily less than 3 $Wm^{-2}$. The peaks of nearly all the error distributions for the different surface types and CODs are around 0, suggesting little systematic bias in the system overall associated with the different surface types and CODs. The mean error for the land distribution is slightly larger at -5.5 $Wm^{-2}$, suggesting a slight negative bias over land. We suspect that this is related to the lack of information about the land-based surface type in the system. If the NN is primarily trained on dark land surfaces, but it is applied to brighter-than-normal land surfaces (e.g. snow- and ice-covered land), the NN will predict lower upwelling SWF than is actually seen by CERES. When excluding data from April and May from this analysis, the mean error for the over-land data is much smaller, supporting our hypothesis that the slight negative shift in the land-based error distribution is related to the land surface brightness that is unaccounted for in this system. Given that the majority of the smoke events analyzed in the study occurred in the summer months (June – August), we do not expect this potential low bias of the NN over bright land surfaces to significantly impact the results of our study.

To further test for biases in the system, we expanded the validation dataset with an additional 25 randomly-selected aerosol-free OMI swaths from the 2005 – 2020 boreal summer data set, bringing the total number of validation pixels up to **300,000 pixels**. When including these additional swaths, the results of the comparison were nearly identical to the results when using the original 50 validation swaths, further suggesting that the system does not contain significant systematic bias.

We have added the following text and figure to the paper:

**3.2 "Validation of the NN against CERES**

Once trained, the NN was first applied to the 50 reserved aerosol-free validation swaths (independent from the 131 aerosol swaths) to validate the NN output against CERES observations. The 50 validation swaths contained about 200,000 pixels to use for validation; we note that similar validation results were obtained when increasing the size of the validation dataset to about 300,000 pixels by adding 25 additional aerosol-free OMI swaths (and co-located MODIS, SSMIS, and CERES data) randomly chosen from the 2005 – 2020 boreal summer study period. Errors were calculated between the NN-estimated aerosol-free SWF and the associated CERES TOA SWF observations, and the distribution of the errors from the 50 validation swaths is shown in Fig. 6a. The error distribution peaks at about 0 $Wm^{-2}$, suggesting little overall bias in the NN-estimated aerosol-free SWF values. To further test for systematic biases in the NN-estimated aerosol-free SWF, we binned the validation dataset first by the SSMIS SIC and surface type, and then by MODIS COD. The NN error distributions binned by the SSMIS surface type and the MODIS COD are shown in Fig. 6b and Fig. 6c, respectively. We found that the mean SWF errors for the error distributions binned by SSMIS SIC and MODIS COD are largely small, with magnitudes primarily less than 3 $Wm^{-2}$. The peaks of nearly all the error distributions for the different surface types and CODs are around 0 $Wm^{-2}$, suggesting little systematic bias in the system associated with the different surface types and CODs. The mean error for the land distribution (Fig. 6b, brown) is slightly larger at -5.5 $Wm^{-2}$, suggesting a slight negative bias over land. We suspect that this is related to the lack of information about the land-based surface type in the system. If the NN is primarily trained on dark land surfaces, but it is applied to brighter-than-normal land surfaces (e.g. snow- and ice-covered land), the NN will predict lower upwelling SWF than is observed by CERES. When excluding data from April and May from this analysis, the mean error for the over-land data is much smaller, supporting our hypothesis that the slight negative shift in the land-based error distribution is related to the land surface brightness that is unaccounted for in this system. Given that the majority of the smoke events analyzed in the study occurred in the summer months (June – August), we do not expect this potential low bias of the NN over bright land surfaces to significantly impact the results of our study."

[Figure]

**Figure 6.** a) Distribution of errors in the neural network (NN)-estimated aerosol-free shortwave flux (SWF) relative to CERES TOA upwelling SWF observations for the 50 validation swaths reserved from the NN training dataset. b) As in (a), but with the errors binned by the SSMIS sea ice concentration (SIC) and surface type. c) As in (a), but with the errors binned by MODIS cloud optical depth (COD).

4. *Figure 5 is not very informative for understanding the neural network architecture. A clearer schematic illustrating the network structure and flow is recommended.*

*RESPONSE*: We thank the reviewer for the recommendation. We have remade Fig. 5 to more clearly illustrate the neural network structure, as shown below, and have inserted this figure into the paper in place of the old version.

[Figure]

**Figure 5.** Architecture of the neural network for estimating L2 aerosol-free SWF from L2 input values of solar zenith angle (SZA), viewing zenith angle (VZA), sea ice concentration (SIC), 2.1 μm reflectance (CH7), cloud optical depth (COD), cloud top pressure (CTP), and surface albedo (ALB). Green circles represent nodes in the input layer, gray circles represent nodes in the hidden layers, and the red circle represents the node in the output layer. All nodes in the neural network are fully connected to the nodes in the next layer, as illustrated by the lines connecting the circles.

5. *The method used for trend estimation should be described in detail. Was data uncertainty incorporated into the trend analysis? In Section 4.3, the error analysis is not pixel-based—how representative is this approach?*

*RESPONSE:* We thank the reviewer for the comments and questions. Our methods for trend estimation, in addition to the methods by which we quantify trend error, are discussed in Section 4.4. Daily ADRF estimates were derived from the daily averaged OMI UVAI value and the ADRF look-up table. Then, a randomly-generated error value matching the error distribution derived in Section 4.3 was added to the derived daily ADRF estimate. Then, all daily ADRF estimates were averaged into monthly values at each grid point. To estimate the trend at each grid point, linear regression was applied to the time series of monthly averaged ADRF estimates, and the difference between the end and beginning points of the fitted trend line represent the trend. This process was repeated 600 times, through a stochastic approach, so that at each grid point there were 600 independent estimates of the monthly ADRF trend from 2005 – 2020.

We account for data uncertainty in this trend approach by perturbing the daily forcing values by an error that matches the distribution from Section 4.3, and through a Monte Carlo method. In that section, we quantified how errors in the input components (i.e. uncertainty in the NN output, uncertainty in applying the LUT to estimate ADRF, uncertainty in the SSMIS SIC / surface type, and uncertainty in the MODIS COD) affect the final daily L3 ADRF estimate. Thus, by conducting 600 independent trend estimates while adding errors that match the combined distribution of the errors from Section 4.3, the spread of the 600 trends at each grid point represents the impacts of data uncertainty on the final trend. We chose to conduct a bulk error analysis rather than a pixel-based error analysis to simplify the error analysis and include more data in each error distribution.

6. *CERES data are used as the reference for shortwave flux (SWF) validation. However, the authors must first assess and validate the accuracy of CERES data specifically over the Arctic.*

   *RESPONSE:* We thank the reviewer for the suggestion. We have modified the end of Section 2.2 to include the following paragraph that describes the validation of CERES data in the Arctic region:

   "CERES data have been used extensively for investigating changes in Arctic radiative energy budgets for both TOA (Duncan et al., 2020; Kay and L'Ecuyer, 2013; Riihelä et al., 2013) as well as the surface (Boeke and Taylor, 2016; Hegyi and Taylor, 2017). Previous studies have also worked to validate Arctic CERES surface radiative fluxes (Di Biagio et al., 2021; Riihelä et al., 2017) and TOA fluxes (Taylor et al., 2022), with the latter seeking to validate CERES TOA radiative fluxes against aircraft-based upwelling radiative flux observations. While Taylor et al. (2022) noted some error in the Arctic CERES Level-2 SSF TOA upwelling SWF resulting largely from errors in the imager-based sea ice concentrations used in the scene classification, the CERES observations compared well overall with the aircraft observations (differences between the CERES and aircraft observations were within 2σ uncertainty). The authors concluded that CERES TOA radiative flux data are suitable for polar climate studies (Taylor et al., 2022)."

7. *On line 564, the authors assume that daily Level 3 ADRF errors are normally distributed. This assumption should be justified with supporting analysis.*

   *RESPONSE:* We thank the reviewer for the comments and suggestions. Our assumption that the daily Level 3 ADRF errors are normally distributed is based on our component-based error analysis we conducted in Section 4.3. A normal distribution provides a good fit for the first two error components, which are the errors in the NN-based L2 ADRF estimates and errors in the application of the ADRF LUT. Though we calculate mean error and error standard deviations for the error distributions arising from the impacts of SSMIS SIC errors and MODIS COD errors on the estimated L3 ADRF values, as shown in Fig. 10 (now Fig. 11), those error distributions are not normally distributed. The vast majority of the errors are equal to 0 because of the size of the SIC and COD bins, so a fitted curve from a normal distribution would likely exhibit a wider spread than is actually present in the error

distribution. However, a normal fit likely provides an overestimate of the spread of the errors for those components. Thus, this overestimate leads to the L3 ADRF errors likely being an overestimate of the true L3 ADRF error. We chose to accept this overestimation of the error to have higher confidence in the trends found to be significant in the study.

[revised manuscript text omitted]